# Hydrodynamic slip can align thin nanoplatelets in shear flow

Catherine Kamal [1], Simon Gravelle[1,2] & Lorenzo Botto[1,3 ✉]

The large-scale processing of nanomaterials such as graphene and $MoS_2$ relies on understanding the flow behaviour of nanometrically-thin platelets suspended in liquids. Here we show, by combining non-equilibrium molecular dynamics and continuum simulations, that rigid nanoplatelets can attain a stable orientation for sufficiently strong flows. Such a stable orientation is in contradiction with the rotational motion predicted by classical colloidal hydrodynamics. This surprising effect is due to hydrodynamic slip at the liquid-solid interface and occurs when the slip length is larger than the platelet thickness; a slip length of a few nanometers may be sufficient to observe alignment. The predictions we developed by examining pure and surface-modified graphene is applicable to different solvent/2D material combinations. The emergence of a fixed orientation in a direction nearly parallel to the flow implies a slip-dependent change in several macroscopic transport properties, with potential impact on applications ranging from functional inks to nanocomposites.

[1] School of Engineering and Material Science, Queen Mary University of London, London, UK. [2] Facultad de Ingeniería y Ciencias, Universidad Adolfo Ibáñez, Viña del Mar, Chile. [3] Process & Energy Department, Faculty of Mechanical, Maritime and Materials Engineering, Delft University of Technology, Delft, The Netherlands. ✉email: l.botto@tudelft.nl

Many inorganic two-dimensional materials have been isolated in the past years, including graphene, Molybdenum disulfide ($MoS_2$) and Boron Nitride (BN)[1]. Made of one or a few layers of atoms, they show physical properties not accessible with bulk materials[2]. In particular, charge and heat transport confined to a plane display unusual behaviour[3], making two-dimensional (2D) appealing candidates for many applications in fields such as electronics[4], energy generation and storage[5], or in biomedicine[6]. But the industrial use of two-dimensional materials requires the understanding of the behaviour of suspended particles in liquids, as some of the most remarkable applications of two-dimensional materials involve their processing with fluids in at least some stages of the production process[7]. Control over the dynamics of suspensions of 2D materials would allow for the development of advanced materials[8], including new-generation nanocomposites[9] and functional inks[10]. In order to predict the behaviour of suspended two-dimensional materials, it is tempting to apply the classical toolbox of colloidal hydrodynamics[11]. But these materials, with their nanometric thickness and unusual interfacial properties[12], challenge the very basic assumptions current colloidal hydrodynamics models rely on. New theoretical tools accounting for the specific properties of 2D nanomaterials are needed.

The current framework for predicting the dynamics of anisotropic colloids in shear flow rests on a mathematical theory due to Jeffery[13]. Developed in 1922, this theory has withstood the test of time and is one of the few theoretical results available for predicting the rheological response of a dispersion of elongated particles[14,15]. Jeffery's theory predicts that a plate-like particle rotates continuously about one of its axis when suspended in a shear field, completing full periodic rotations. This rotational motion is due to the torque exerted on each particle by the shear flow. The rotational dynamics of the suspended particles and the ensuing orientational microstructure affects the value of the suspension viscosity[16,17], and impacts other effective two-phase transport properties, such as thermal and electrical conductivities[18]. Controlling these macroscopic properties is paramount to delivering the promise of two-dimensional materials in market applications.

The study of two-dimensional materials is complicated by their unusual interfacial properties. Recent studies highlight the importance of hydrodynamic slip at the interface between water and an atomically smooth surface[12,19,20], i.e. the ability of fluid molecules to 'slip' on the solid surface rather than 'adhering' to it. The slip over the surface is usually characterised by the so-called slip length $\lambda$: the distance within the solid at which the relative solid-fluid velocity extrapolates to zero[21]. For relatively large objects, the slip length is much smaller than the typical scale of the system and so the no-slip boundary condition holds almost exactly. For a system with a characteristic dimension close in magnitude to the slip length, however, the effect of the slip becomes significant. For example, the rate of flow through nanoporous carbon-based membranes ($\lambda \sim 10$ nm) is enhanced by up to an order of magnitude as compared to classical predictions[22]. Therefore it is natural to reconsider Jeffery's predictions in the context of nanoplatelets with hydrodynamic slip suspended in water. It is currently unclear what effects may arise in suspensions due to slip.

In the context of colloidal hydrodynamics, slip is known to reduce the hydrodynamic stress applied by the shearing liquid on the particle's walls, resulting, e.g., in a slowing down of the rotational dynamics of spheres and infinite cylinders with axis in the vorticity direction[23,24] (Supplementary Note 1). A similar effect has been predicted for elongated particles of moderate aspect ratio ($b/a \sim 0.5$) and small slip length ($\lambda \sim a/10$)[25], as well as for slightly oblate spheroids with $b/a = 5/6$, $\lambda/a \leq 1$, and their

longer axis perpendicular to the plane of the flow[26]. The effect of slip on the hydrodynamic torque and drag of rotating or settling elongated particles in quiescent fluid has also been studied[27–31], as well as for plate-like geometries of relatively large thickness[32]. But nanoplatelets, made for example of carbon, BN, or $MoS_2$, can exhibit extreme aspect ratios (typically, $b/a \sim 10^{-3}$) and can have significant slip lengths, often larger than the nanoplatelet thickness. The effect of slip in such conditions must be reconsidered.

In the present work, using a combination of Molecular Dynamics (MD) and Boundary Integral (BI) simulations, we demonstrate that in the case of graphene in water, slip induces a dramatic change in the rotational behaviour that goes beyond a simple slip-dependent change in rotational velocity. In particular, we show that Jeffery's theory[13], which predicts no stable orientation, fails to describe the rotational dynamics of graphene in the presence of comparatively large slip. In our simulations, the particle attains a stable orientation rather than performing periodic orbits. This unexpected result is due to a unique combination in our system of an extremely small effective thickness of the particle (~0.5 nm for single graphene layer[33]) and a significant slip at the graphene-water interface ($\lambda \geq 10$ nm). The theory relies on a combination of slip and geometry, and is therefore not limited to water and graphene. In addition, using asymptotic methods, we are able to develop a continuum-based theory that accurately predicts the MD data. We also extend our MD results to systems other than pure graphene in water. We first consider a graphene-oxide platelet, with both edge and basal plane oxidation. Results show a transition from a stable orientation regime to a regime in which the particle rotates for a certain degree of basal plane oxidation. In addition, results obtained with non-aqueous solvents are qualitatively consistent with the predicted slip-induced alignment. More broadly, our results suggest that even nanometric slip lengths can change the rotational dynamics of a large class of 2D nanomaterials and solvents.

## Results

**Rotational dynamics.** We perform MD simulations of a freely suspended graphene particle in a shear flow using LAMMPS[34]. In these simulations, the particle is rigid and free to rotate. The platelet consists of a stack of $n$ graphene layers of approximate dimensions 3.4 nm along $\hat{e}_x$, 2.5 nm along $\hat{e}_z$ and separated by a distance equal to 3.35 Å[35]. The thickness of the platelet is $2b$, the length $2a$, and the spanwise dimension of the computational domain in the $\hat{e}_z$ direction is $w$ (Fig. 1). Note that the analysis of such quasi-2D configuration is not restrictive, and the results are valid for geometries that vary in the $\hat{e}_z$ direction (e.g. a disk-like particle) up to a numerical prefactor (see the asymptotic analysis of the hydrodynamic traction for a 3D axisymmetric disk in the Supplementary Note 1). The fluid consists of a number $N = 10^4$ of water molecules, enclosed in the $\hat{e}_y$ direction by two moving walls (Fig. 1), and periodic in the $\hat{e}_x$ and $\hat{e}_z$ directions. We use the TIP4P/2005 model for water[36], and the Amber96 force field for the carbon-oxygen interactions[37]. The slip length of water on a planar graphene surface, which depends on the force fields, is estimated here from Poiseuille flow simulations as $\lambda = (60 \pm 5)$ nm (see Methods). Water molecules are maintained at a constant temperature $T = 300$ K using a Nosé-Hoover temperature thermostat[38,39] applied only to degrees of freedom in the $\hat{e}_y$ and $\hat{e}_z$ directions. The shear flow is produced by the relative translation of the two parallel walls, producing a shear rate $\dot{\gamma} \approx 5 \times 10^{10}$ s$^{-1}$. More details concerning the MD simulations are given in the Methods section.

We let a platelet with $n = 2$ free to rotate around the $\hat{e}_z$ axis (Fig. 1). While we were expecting a rotation in the direction of the shear as predicted by Jeffery theory[13], we observe that the platelet

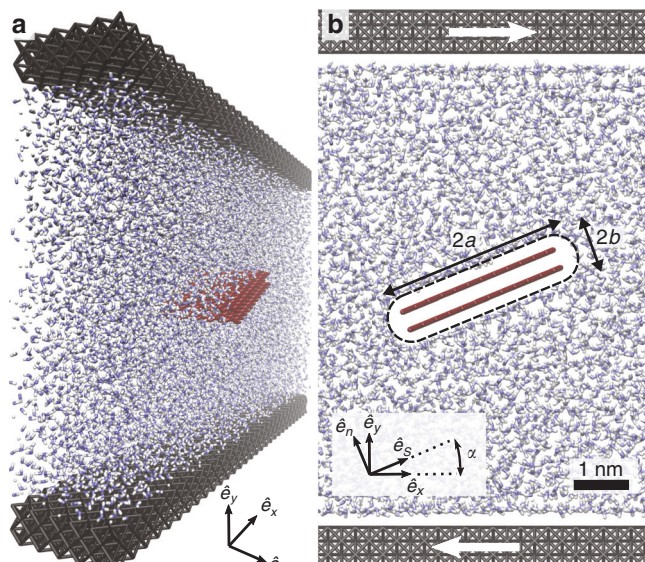

**Fig. 1 View of the MD system. a** Perspective view of a graphene bilayer in a shear flow of strength $\dot{\gamma}$, as extracted from MD simulations[80]. The blue and white dots are water molecules, and the black layers at the top and bottom are the shearing walls. **b** Zoom on the graphene bilayer, inclined by an angle $\alpha$ with respect to the flow. Comparison with continuum simulations is made by calculating the stress on the continuous surface represented by the dashed outline.

rotates in the opposite direction, and reaches a time-average equilibrium angle $\alpha_c \approx 20°$ (Fig. 2a; Supplementary Movie). The platelet oscillates around $\alpha_c$ due to Brownian fluctuations. A similar simulation for a monolayer graphene platelet gives a slightly smaller time-average equilibrium angle, $\alpha_c \approx 18°$. Because the rotational Peclet number characterising the ratio of viscous to Brownian forces is much larger than 1 (we calculate $Pe = \dot{\gamma}/D_r \approx 100$, where $D_r$ is the rotational diffusion coefficient for a disk of radius $a$; $D_r \approx 3k_BT/(32\eta a^3)$, where $k_B$ is the Boltzmann constant and $T$ the temperature[16]), the stable equilibrium angle is associated with the hydrodynamic stress distribution over the platelet surface and its moment, the hydrodynamic torque. In contrast, MD simulations of a platelet presenting a no-slip surface produce orbits similar to those predicted by Jeffery (Fig. 2b).

**Hydrodynamic traction and torque**. The key to understanding the rotational dynamics of a particle in a Stokes flow is the calculation of the hydrodynamic torque $T$ exerted by the fluid on a fixed particle[40]. Using MD, we fix the platelet's orientation at a specific angle $\alpha$ and measure the hydrodynamic torque $T$ exerted on it (Fig. 3). The sign of the torque determines whether the platelet, when allowed to rotate, will rotate clockwise (for $T < 0$), counter-clockwise (for $T > 0$), or reach a stable orientation (for $T = 0$). The data shows clearly a transition from $T > 0$ to $T < 0$ for a critical angle, which for a bilayer is $\alpha_c \approx 22°$ and for a monolayer is $\alpha_c \approx 18°$, in good agreement with the dynamic simulations. The blue continuous curve in Fig. 3 shows the prediction of Jeffery for no-slip platelets. In this case, the torque is negative for any value of the inclination angle.

In order to understand why the torque changes sign for $\alpha < \alpha_c$, we have calculated the hydrodynamic stress distribution over the particle surface when $\alpha = 0$ (Fig. 4a, b). We compare the MD results with a continuum resolution of the Stokes equation for an incompressible fluid using the BI method. The BI method is known to be a particularly accurate alternative to other numerical

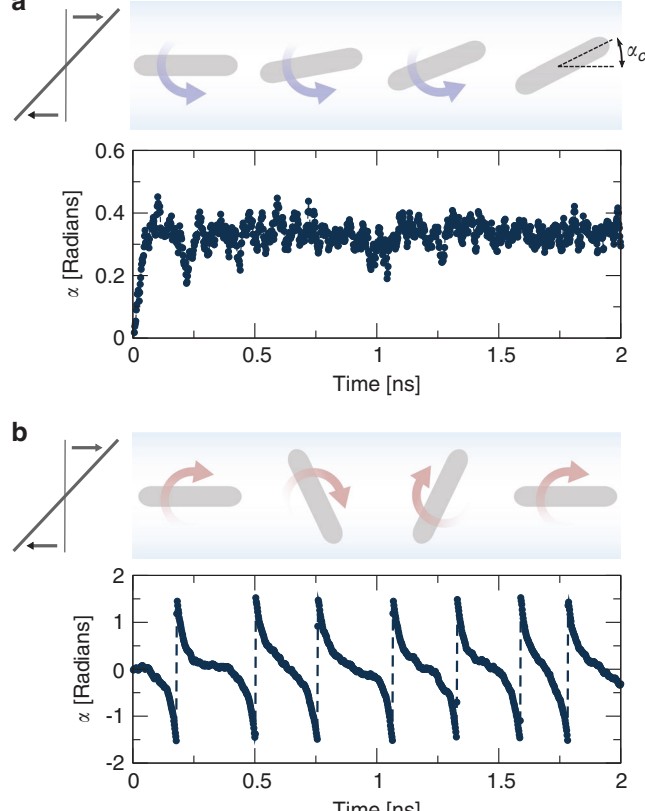

**Fig. 2 Effect of slip platelet dynamics. a** Top: according to our theory, a platelet with large hydrodynamic slip initially aligned with the flow will rotate in the opposite direction from that of the vorticity of the undisturbed flow and towards a stable inclination angle $\alpha_c$. Bottom: time evolution of orientation angle $\alpha$ from dynamic MD simulation of a freely suspended graphene bilayer, with half length $a = 1.7$ nm, aspect ratio $b/a = 0.25$, and slip length $\lambda = (60 \pm 5)$ nm. **b** Top: according to Jeffery's theory, a platelet initially aligned with the flow will rotate continuously and in the same direction as the vorticity of the flow. Bottom: orientation angle $\alpha$ from MD simulation of a freely suspended no-slip platelet, with half length $a = 1.8$ nm, aspect ratio $b/a = 0.2$ and slip length $\lambda \approx 0$.

methods for solving the Stokes equation, because it requires implicitly solving an integral over the platelet surface instead of the full domain[41]. A Navier slip boundary condition is assumed at the graphene surface, with a slip length $\lambda$ (see Methods, Eq. (10)). The hydrodynamic stress is evaluated at a reference surface consisting of a rectangular parallelepiped of length $2a$, thickness $2b$, and presenting rounded edges corresponding to the smoothing of the molecular flow by the edges (dashed line in Fig. 1b). Analysis of the MD density profiles suggests that this surface gives an optimal approximation, from a hydrodynamic prospective, of the 'true' graphene surface. More details concerning BI calculations are given in the Methods section, as well as in the Supplementary Method. The torque calculated using BI with a slip length $\lambda = 60$ nm, comparable with the MD value measured for a planar graphene surface, gives an excellent agreement with MD simulations (Fig. 3). The excellent agreement between the MD and BI calculations suggests that atomistic hydrodynamic features, such as fluid structuring near the surface[12], or non-uniformities in surface properties leading to differences in slip length between the edges and the flat surfaces[42], do not induce leading order effects on the torque. Therefore, a continuum formulation based on the solution of the Stokes equation with a

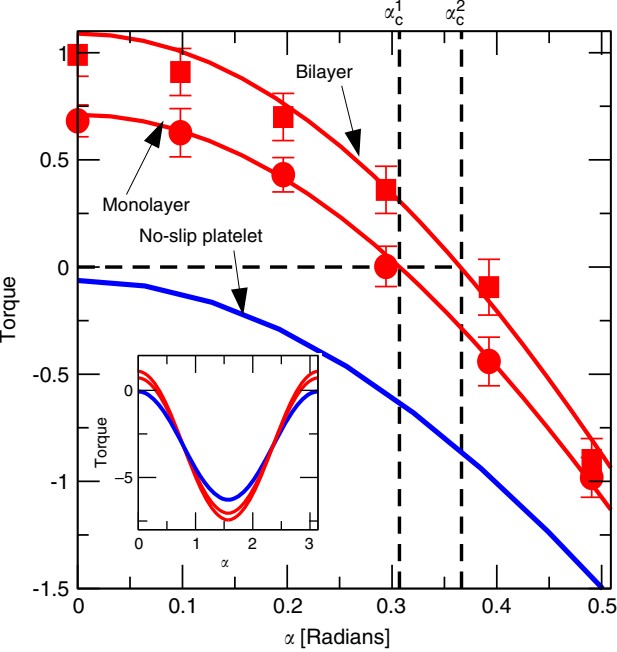

**Fig. 3 Torque versus platelet orientation.** Non-dimensional hydrodynamic torque $T/(a^2 w\dot{\gamma}\eta)$ versus platelet inclination angle $\alpha$. Symbols correspond to MD simulations for a monolayer with aspect ratio $b/a = 0.15$, half length $a = 1.7$ nm and slip length $\lambda = (60 \pm 5)$ nm (disks) and a bilayer with aspect ratio $b/a = 0.25$, half length $a = 1.7$ nm and slip length $\lambda = (60 \pm 5)$ nm (squares); red lines are BI calculations for the corresponding values of $\lambda/b$ and $b/a$. Error bars correspond to the standard deviation from 10 simulations of 0.4 ns. The dashed lines mark the inclination angle $\alpha_c$ for which the hydrodynamic torque is zero. The blue line corresponds to BI calculations in the case of a monolayer with no-slip boundary condition. Insert: Torque vs platelet inclination angle for $\alpha \in [0, \pi]$.

single slip parameter can be used for predicting the stress applied on a graphene nanoplatelet.

The torque can be expressed as (Supplementary Method)

$$T_\lambda(\alpha) = \int_{-a}^{a} \left( s\bar{f}_n - h(s)\Delta f_s \right) dS, \qquad (1)$$

where $\bar{f}_n$ and $\Delta f_s$ are the average hydrodynamic traction acting in the direction normal and tangential to the platelet respectively, $s$ is the coordinate running along the centreline of the platelet, $h(s)$ is the distance from the reference surface to the platelet's centreline, and $dS$ is the element of surface area. We measure $\bar{f}_n$ and $\Delta f_s$ as function of $s$, with $s = 0$ in the centre of the platelet and $s = \pm a$ corresponding to the edges (Fig. 4a, b). The stress distribution displays two distinct regions: a region near the edges characterised by sharp peaks in both normal and tangential tractions, and a region far from the edges where both normal and tangential stresses are comparatively small. The most notable effect of slip is a large reduction in the tangential stress along the flat surface of the graphene layer (Fig. 4b). If the platelet surface was a no-slip surface, we would expect $\Delta f_s(s) \simeq \eta\dot{\gamma}$ at the flat surfaces of the platelet. Results show instead $\Delta f_s \ll \eta\dot{\gamma}$ in this region as a result of the slippage of the water molecules at the graphene-water interface.

The observed dynamics can be understood from simple arguments, following a thorough analysis of Eq. (1). Let us call $T_n$ the torque due to the normal traction (left integrand term in Eq. (1)), and $T_s$ the torque due to the tangential traction (right integrand term in Eq. (1)). When the particle is aligned with the flow ($\alpha = 0$), because the tangential stress $h(s)\Delta f_s$ at the surface is

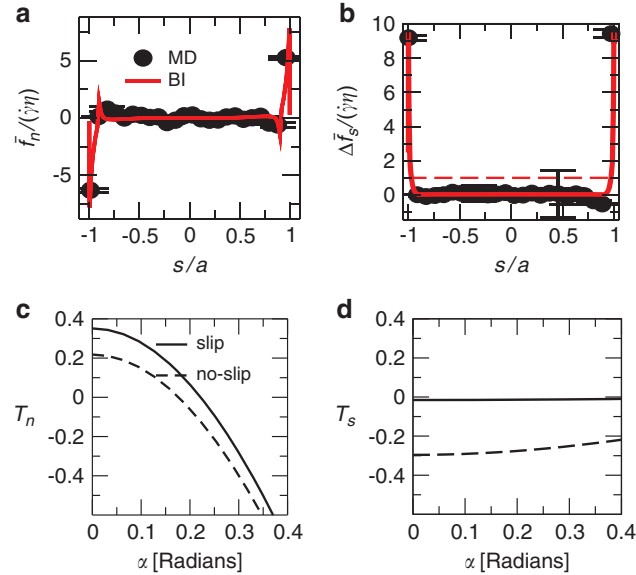

**Fig. 4 Normal and tangential traction and their contribution to the torque. a, b** Non-dimensional hydrodynamic traction normal (**a**) and tangent (**b**) to the surface of a graphene platelet for $\alpha = 0$ and $n = 2$, comparing MD and BI simulations. Red dashed line is the no slip estimate $\Delta f_s/(\eta\dot{\gamma}) \cong 1$. Error bars correspond to standard deviations in MD measurements. **c** Torque component corresponding to normal traction $(a^2 w\dot{\gamma}\eta)T_n = \int_S s\bar{f}_n dS$ (left integrand term of Eq. (1)) as a function of the orientation angle $\alpha$ for a no-slip (dashed line) and slip (full line) boundary condition. **d** Torque component corresponding to tangential traction $(a^2 w\dot{\gamma}\eta)T_s = -\int_S h(s)\Delta f_s dS$ (right integrand term of Eq. (1)).

reduced due to slip, $T_s$ decreases by about one order of magnitude from the no-slip value (Fig. 4d). But $T_n$ in presence of slip decreases only by a factor of ~2 from the no-slip value (Fig. 4c). This can be explained from the observation that the main contribution to $T_n$ comes from the stress peaks near the edges; at the edges, the normal stress originates from the reorientation of the streamlines due to the non-penetration boundary condition, and this effect is independent of $\lambda$. As a result of $T_n > T_s$, the total torque on the platelet for $\alpha = 0$ becomes positive (counter clockwise) for a sufficiently large slip length (Fig. 5). On the other hand, the direction of rotation when the particle is oriented normally to the flow ($\alpha = \pi/2$) is clockwise regardless of the value of $\lambda$ (Fig. 5). Hence the particle finds an equilibrium orientation at an intermediate value of $\alpha$.

It could be expected that a small amount of slippage would just slow down the dynamics with respect to what is predicted by Jeffery's theory[25]. Our results instead demonstrate that the presence of even relatively small slip can qualitatively change the rotational dynamics of the platelet by perturbing the balance of tangential and normal torques (Fig. 5).

In the next section, we perform an asymptotic analysis of the BI equations in the limit $a \gg b$, and predict the value of the minimum slip length $\lambda_c$ needed for the rotational dynamics to change character. We also predict the value of the critical angle $\alpha_c$ as a function of the slip length $\lambda$ and platelet dimensions.

**Critical slip length and estimate of $\alpha_c$.** Before we analyse the value of the minimal slip length for stable orientation, we consider how the critical angle $\alpha_c$ relates to the hydrodynamic torque. The value of $T_\lambda$ for a given value of $\alpha$ can be calculated from the torque values at $\alpha = 0$ and $\alpha = \pi/2$ according to

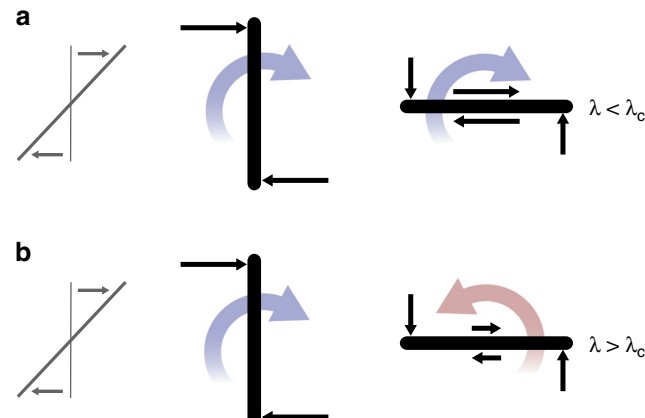

**Fig. 5 Dominant contributions to the torque.** Schematic of the dominant contributions to the torque applied on the platelet under shear flow for small slip length (**a**) and large slip length (**b**). The coloured arrows indicate the direction of rotation.

$T_\lambda(\alpha) = T_\lambda(0)\cos^2\alpha + T_\lambda(\pi/2)\sin^2\alpha$[40]. Setting $T_\lambda(\alpha_c) = 0$, we get

$$\tan \alpha_c = \sqrt{\frac{-T_\lambda(0)}{T_\lambda(\pi/2)}}. \quad (2)$$

For $\alpha = \pi/2$, the slip length $\lambda$ has a negligible influence on $T_\lambda$ because the incoming velocity is directed almost perpendicular to the surface of the platelet, except at the edges. Thus, in analogy with the no-slip case,

$$T_\lambda(\pi/2) \approx -w\eta\dot{\gamma}c_1 a^2, \quad (3)$$

where $c_1$ is a positive constant. From BI calculations, $c_1$ is found to be almost independent of the particle aspect ratio $a/b$, and approximately equal to $c_1 \simeq 6.6$ for both $n = 1$ and $n = 2$ (inset of Fig. 6).

In contrast, $T_\lambda(0)$ depends strongly on $\lambda$ (Fig. 4c, d and inset of Fig. 6). We have quantified the contributions $s\bar{f}_n$ and $h\Delta f_s$ by asymptotic expansion of these stress components in powers of $b/a$. Equating equal-order terms in the BI formulation, one can calculate the surface traction to different orders of approximation (Supplementary Note 1). We find that because $f_n$ acts perpendicularly to the boundary surface, the contribution to the torque from $s\bar{f}_n$ is independent of $\lambda$ to leading order. The corresponding torque contribution is positive (counter-clockwise) and scales as $\int s\bar{f}_n dS \propto ab$[43]. In contrast, the tangential traction depends strongly on $\lambda$, approximately as

$$\Delta f_s \approx \frac{\eta\dot{\gamma}}{1 + 4\lambda/(\pi a)}, \quad (4)$$

far from the edges. The average value of $\Delta f_s$ along the top surface of a bilayer as extracted from MD is $\Delta f_s/\dot{\gamma}(\eta) \approx 0.05$. This value compares well to the value of $\Delta f_s/\dot{\gamma}(\eta) = 0.06$ given by Eq. (4). The corresponding contribution to the torque is $\int -h\bar{f}_s dS = abw\Delta f_s$. For a no-slip surface, the exact cancellation between the torque contributions due to tangential and normal tractions to $\mathcal{O}(b/a)$ gives rise to Jeffery's dynamics[43]. In the presence of slip, the negative contribution to the torque from tangential stresses decreases with $\lambda$, while the positive contribution from normal stresses is independent of $\lambda$. Hence, the aforementioned cancellation of torque contributions is incomplete. The result of this incomplete cancellation is that the torque will change sign at a critical value $\lambda_c$ of the slip length (Fig. 6).

The value of $\lambda_c$ can be calculated by equating the difference between the torque contributions from the $\mathcal{O}(b/a)$ tangential and

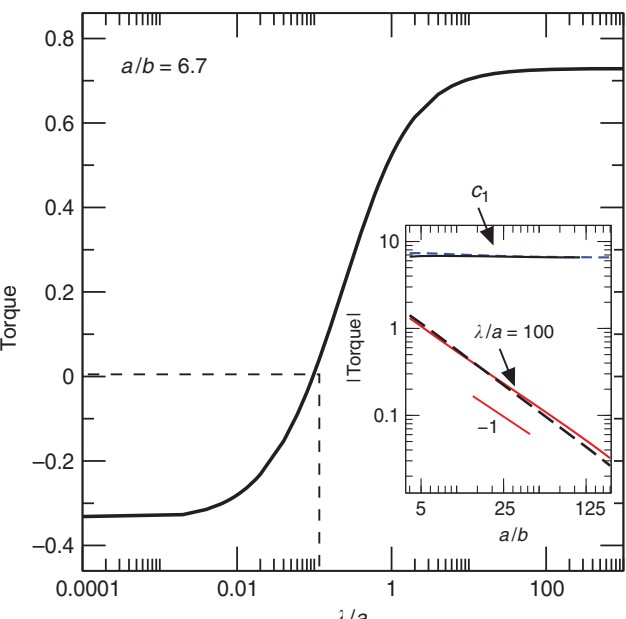

**Fig. 6 Torque versus hydrodynamic slip.** Non-dimensional hydrodynamic torque $T/(a^2 w\dot{\gamma}\eta)$ vs slip length $\lambda/a$ for $\alpha = 0$ and $a/b = 6.7$, based on full BI simulations. Dashed line marks the slip length when $T$ changes sign. Inset: non-dimensional hydrodynamic torque vs aspect ratio for (i) a monolayer at $\alpha = \pi/2$ ($c_1$) when $\lambda = 0$ (full line) and $\lambda/a = 100$ (dashed line), and (ii) a monolayer (full red lines) and a bilayer (black-dashed lines) at $\alpha = 0$ when $\lambda/a = 100$.

normal stresses to the $\mathcal{O}(b^2/a^2)$ torque term. This calculation reveals that, to leading order, $\lambda_c$ is of the order of the thickness of the nanoplatelet, independently of the length of the particle (Fig. 7). This result is counterintuitive because $\lambda_c$ is related to the torque and the torque does depend strongly on $a$. The fact that $\lambda_c$ is approximately independent of $a$ means that we can extrapolate our MD results to realistic values of the nanoplatelet length. In terms of orders of magnitude, the criterion for stable orientation is

$$\lambda > \lambda_c, \text{ where } \lambda_c \sim b. \quad (5)$$

A brief quantitative explanation for Eq. (5) is the following. Let's consider the leading torque contributions from $\bar{f}_n$ and $\Delta f_s$ (far from the edges) when $\lambda/a \ll 1$. For the torque to change sign, the contribution to the torque from the normal traction must be larger than the corresponding contribution from the tangential traction. As anticipated before, when $\lambda = 0$, these two torque contributions cancel each other exactly to leading order. The clockwise torque predicted by Jeffery originates from the second-order torque term $\Delta f_s^{(2)} \sim b/a$[43]. For $\lambda/a \ll 1$, $\Delta f_s^{(1)} \approx \eta\dot{\gamma}(1 - 4\lambda/(\pi a))$ (Eq. (4)). Since $\bar{f}_n$ is independent of $\lambda$ to $O(b/a)$, the leading order cancellation due to $\bar{f}_n$ is the same as for $\lambda = 0$, so that the remaining hydrodynamic torque comes from $-\eta\dot{\gamma}(4\lambda/(a\pi)) + \Delta f_s^{(2)}$. For the torque to change sign, $\Delta f_s^{(2)} \sim \eta\dot{\gamma}b/a$ must be smaller than $\eta\dot{\gamma}4\lambda/(a\pi)$. Therefore the critical value of $\lambda$ is of the order of $b$. This analysis can be repeated for objects of finite extent in the $\hat{e}_z$ direction (e.g. a disk-shaped particle) and leads to similar results. Since the analysis only requires the platelet to be 'thin', the result also holds for a variety of plate-like shapes, such as a thin particle with an elliptical cross section. The only difference is that one has to account for a numerical prefactor that depends on the specific geometry of the object (Supplementary Note 1).

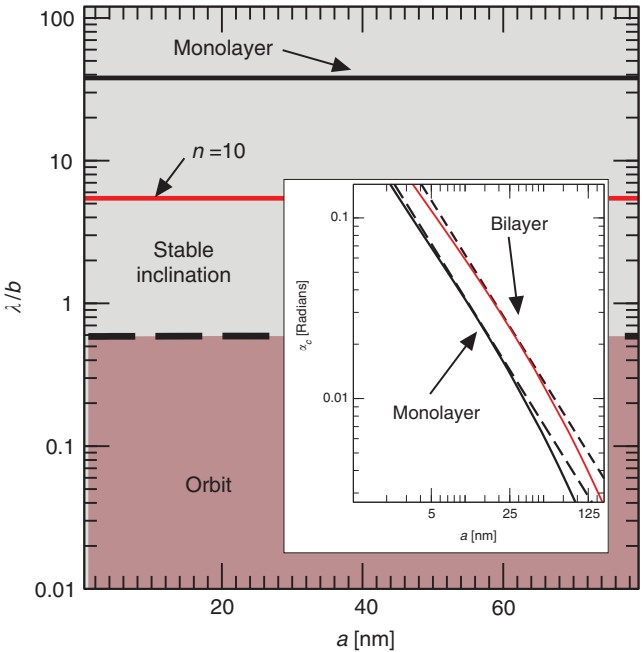

**Fig. 7 Stability phase diagram.** Regimes of rotational dynamics (periodic orbit as predicted by Jeffery's orbit, or stable inclination) as a function of the slip length and aspect ratio for a graphene monolayer. Full lines are $\lambda/b$ for $\lambda = 10$ nm and: $n = 1$ (black line); $n = 10$ (red line). These platelets are well within the region of 'stable inclination'. The simulations suggest that the critical slip length separating the two regions is comparable to $\lambda_c \sim b$. Inset: Critical inclination angle $\alpha_c$ vs length of nanoplatelets for a monolayer ($n = 1$) (black line) and bilayer ($n = 2$) (red line) with a fixed slip length $\lambda = 60$ nm. Dashed (black) line is the estimate $c\sqrt{b/a}$.

An upper bound for $\alpha_c$ can be obtained as follows. In the limit $\lambda/a \rightarrow \infty$, $T_\lambda(0)$ approaches a maximum value (Fig. 6). In this limit, the counter-clockwise torque from the edge dominates and

$$\lim_{\lambda/a \rightarrow \infty} T_\lambda(0) \approx \int s\bar{f}_n dS \approx c_2 wa^2(b/a), \quad (6)$$

where $c_2$ is a prefactor that depends on $n$ (Fig. 6 suggests $c_2 \approx 3.71$ for $n = 1$ and $c_2 \approx 3.91$ for $n = 2$). Because $T_\lambda(\pi/2)$ remains almost constant as $\lambda/a \rightarrow \infty$, Eqs. (3) and (2) yield

$$\alpha_c \lesssim c\sqrt{b/a}, \quad (7)$$

where $c = \sqrt{c_2/c_1}$ is an $\mathcal{O}(1)$ constant ($c \approx 0.75$ for $c_1 \approx 6.6$ and $c_2 \approx 3.71$). For example, comparing to the MD simulations given in Fig. 3 with $\lambda/a \approx 35$, Eq. (7) gives $\alpha_c \approx 17°$ for $n = 1$ and $\alpha_c \approx 22°$ for $n = 2$. As $\lambda/a$ increases the magnitude of $T_\lambda(0)$ decreases (Fig. 6), causing $\alpha_c$ to become smaller than $c\sqrt{b/a}$ (Eq. (7)). The value of $\alpha_c$ shown in Fig. 7 shows a rather small deviation as $\lambda/a$ decreases, suggesting that $c\sqrt{b/a}$ compares well to the actual value of $\alpha_c$ even when $\lambda/a \sim \mathcal{O}(1)$. For $\lambda/a \ll 1$, the inequality will still hold (possibly with a different scaling of $\alpha_c$ with respect to $b/a$), and the platelet will be practically aligned with the shear flow.

**Dispersability and surface modification.** So far, we have focused on pure graphene in water because of the quality and quantity of available data[44]. It may be argued that pure graphene is not easily dispersable in water. However, there are several possibilities to obtain a stable dispersion of graphene in water, such as modifying the graphene surface[45–50], or adding dispersants to the solution[51,52]. In these situations, our theory is expected to apply as long as the slip length remains larger than the platelet's

thickness. Increasing graphene's dispersibility will not necessarily alter the relevant hydrodynamic stress distribution at the particle surface. For example a selective modification of the edges of a graphene platelet can lead to an improved dispersibility[49,50] without altering the slip at the basal surface.

To illustrate the effect of edge-selective modification on the orientation of a graphene nanoplatelet in water, we have performed additional MD simulations using a monolayer graphene platelet presenting edge-selective oxidation (see Methods). The results show no significant change in the time-average equilibrium angle $\alpha_c$ as compared with pure graphene (Fig. 8a). This is inline with our theory which shows that the contribution to the torque from the edges is independent of $\lambda$ to leading order. Thus, the validity of our predictions is not undermined by edge modification, and our theory should apply to cases in which particle aggregation is prevented by modification of the edges (including inducing charges at these locations[46,47]).

Additionally, we have performed MD simulations using a monolayer graphene platelet with oxidation at the basal plane and at the edges. The largest the degree of oxidation at the basal plane, the smallest the slip length (see Methods, Table 1)[53]. Therefore, increasing the degree of oxidation impacts the particle orientation: as the slip length $\lambda$ decreases (but remains larger than the particle half width $b$), the average angle $\alpha_c$ decreases, in good agreement with our theory (Fig. 8a). When $\lambda$ becomes comparable with $b$, a smooth transition occurs and the particle completes occasional rotations with time period $P$ (Fig. 8b). As $\lambda/b \rightarrow 0$, $P$ eventually decreases toward the value of 0.25 ns predicted by Jeffery's theory[13].

Alternatively, graphene forms stable dispersion without requiring alterations to the surface chemistry in several solvents, for example in *N*-Methyl-2-pyrrolidone (NMP) or in cyclopentanone (CPO)[7,54]. We have performed MD simulations of freely suspended graphene particle using either NMP or CPO (see Methods). In both cases, a graphene platelet aligns at a small angle $\alpha_c$, while a no-slip platelet rotates as predicted from Jeffery theory (Fig. 8). Note that the measured values of $\alpha_c$ in these solvents are slightly lower than in water.

Finally, these results should apply to solid materials other than graphene. For example, the slip length predicted for water in contact with hexagonal boron nitride is ~3.3 nm[12], a value roughly one order of magnitude larger than the typical thickness of a boron nitride nanoplatelet. Beyond 2D nanomaterials, large slip length have been reported on hydrogel surfaces[55]. Relatively large slip lengths can be obtained with conventional materials by using surface modification[56], depletion layers[57], or surface nanobubbles[58], opening up opportunities for the experimental verification of our theory using mesoscale colloidal objects in simple or complex fluids.

## Discussion

We have demonstrated that in the presence of hydrodynamic slip effects, there exists a regime in which a rigid nanoplatelet suspended in a liquid does not rotate when subject to a shearing flow. Instead, the particle is trapped in a 'hydrodynamic potential well' at a small inclination angle $\alpha_c$ with respect to the flow direction. We found an excellent agreement between molecular dynamics for a graphene-water system and a continuum theory based on a boundary integral formulation that includes hydrodynamic slip.

The main result of our numerical and asymptotic analysis is that a stable orientation occurs when the hydrodynamic slip length $\lambda$ is larger than the thickness of the platelet (Fig. 7). Our theory also predicts that the contribution to the torque from the edges is independent of $\lambda$ to leading order, suggesting that a

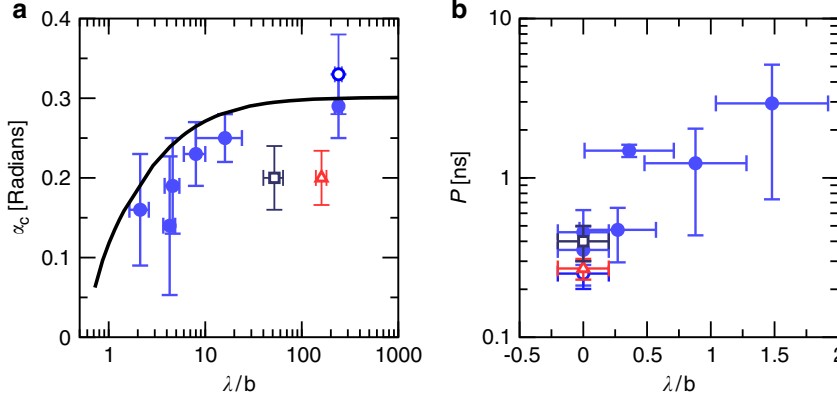

**Fig. 8 Impact of the slip length on the platelet dynamics. a** Average angle $\alpha_c$ as a function of $\lambda/b$ for a monolayer graphene platelet $a = 1.7$ nm. Data corresponds to pure graphene in water (open disk), pure graphene in NMP (open square), pure graphene in CPO (open triangle) and surface-modified graphene in water (full disks). Error bars on both $\alpha_c$ and $\lambda$ correspond to standard deviations in MD measurements. **b** Average time period $P$ of the orbit, as a function of $\lambda/b$. Data corresponds to a no-slip platelet in water (open disk), no-slip platelet in NMP (open square), no-slip platelet in CPO (open triangle) and surface-modified graphene in water (full disks). Error bars on $P$ are the standard deviation measured on several rotations.

modification of the platelet's edges has a negligible effect on the occurrence of a stable orientation, as proved by MD simulations of edge-oxidised graphene platelet (Fig. 8). The effect of surface modification of the basal plane, however, must be assessed more critically. We showed that an increase in the degree of oxidation at the basal plane eventually alters the stable orientation and leads to a transition toward continuum rotation of the particle (Fig. 8), in agreement with our theory which predicts a transition for $\lambda \sim b$.

The theory represented here is based on the assumption that the nanoplatelet is rigid. For $\alpha = 0$, the platelet will behave as a rigid object provided that the viscous forces $\sim \eta \dot{\gamma} ab$ are much smaller than the bending forces $\sim B/a$, where $B \sim Db^3$ is the bending rigidity, and $D \approx 10^{11}$ J m$^{-3}$ (ref. [59]; Supplementary Note 2). A criterion for the onset of deformability effects can be obtained by setting $\eta \dot{\gamma}(a/b)^2/D = 1$. For a typical shear rate $\dot{\gamma} = 10^4$ s$^{-1}$, and a solvent with viscosity $\eta \sim 10^{-3}$ Pa s, any nanoplatelet with aspect ratio $b/a < 10^{-5}$ will thus appear rigid.

The effect of Brownian forces is to randomise the orientation. Provided that the angular dispersion about $\alpha_c$ caused by Brownian forces is sufficiently small, our large-Peclet number theory provides a prediction of the time-average angle the particle will oscillate about, and is a starting point for predicting the full orientational particle distribution function for platelets in the presence of slip. Therefore, our results have implications for the rheology of graphene dispersions for a wide range of Peclet numbers.

The importance of our result stems from the fact that changes in the orientation distribution of a particle will affect all the (effective) two-phase transport properties of a liquid dispersion (e.g. the effective viscosity of nano-inks, or the heat/mass transfer coefficients of nanofluids)[60]. It is clear that the transport properties of a mixture where the particles are aligned in an average sense with the flow are considerably different from the ones obtained when the particles are aligned with the flow at each instant. For instance, an atomically thin plate-like particle that is aligned instantaneously with the streamlines will disturb the flow very little, resulting in a smaller distortion of the streamlines, a smaller viscous dissipation, and hence a smaller suspension's viscosity than if the particle was rotating[16]. Such decrease in viscosity, which could be relevant for improving the flowability of nano-inks, could be measured experimentally as a way to evidence slip effects (a similar suggestion was made in Kroupa (2017)[24] for a concentrated dispersion of spherical particles). Another example of application of our findings is coating for gas barrier, where it is desirable to obtain a distribution of plate-like

particles aligned with the boundary, so as to lead to a longer gas diffusion path[61]. We are suggesting that slip combined with shear can enable to achieve this objective.

Besides complex liquids, our result has implications for two-phase solid. In materials processing methods that involve the "solidification" of a continuous liquid phase containing nanoplatelets (as in the processing of polymer nanocomposites[62]), the change in microstructure of the liquid suspension will be inherited by the solid. In these applications, to obtain superior mechanical properties it is usually desirable to have almost complete alignment of the platelets[63]. Our results suggest that in the high-Peclet number limit all plate-like particles with $\lambda > b$ will be nearly aligned with the flow, while for $\lambda < b$ a larger variance is expected.

The validity of the theory discussed here could be tested by measuring experimental observables that are sensitive to second-order statistical moments, such as the 'degree of orientation'. The 'degree of orientation' of the particles, can be assessed by rheo-optics experiments[64–66]. Contrarily, the average particles orientation angle may not be ideally suitable for discriminating between rotating and aligned particles because highly elongated plate-like particles are expected to align with the flow in a time-average sense regardless of the hydrodynamic slip[16,67].

By challenging Jeffery's theory, whose presence is pervasive in the theory of anisotropic colloids, our results offer an important new direction of research for the hydrodynamics of colloidal systems. Our work demonstrates that even nanometric slip lengths can lead to drastic changes in particle dynamics, hence suggesting that slip can be used to tune the orientational microstructure in suspensions of anisotropic particles, with important implications for rheology and the development of new-generation anisotropic materials.

## Methods

**Molecular dynamics simulation**. All simulations are performed with LAMMPS[34]. The simulation box is typically 14 nm along $\hat{e}_x$, 13 nm along $\hat{e}_y$ and 2.5 nm along $\hat{e}_z$. A platelet, either made of pure graphene, of graphene-oxide, or of no-slip material, is immersed in a solvent (Fig. 1 of the main text). In the case of a multilayer graphene platelet, the distance between two layers of graphene is chosen to be equal to the experimental value (3.35 Å[35]). Moving walls are used to enclose the fluid in the $\hat{e}_y$ direction. Following the work of Huang et al.[68], for the atoms of the moving walls, LJ parameters are chosen to create a physically reasonable, idealised surface, with $\sigma_{ww} = 3.374$ Å, where the index 'w' stand for 'wall', and a close-packed density of $\rho_w = \sigma_{ww}^{-3}$. Also following Huang et al, we choose $\epsilon_{ww} = 2.084$ kcal mol$^{-1}$ to create a hydrophilic surface characterised by a contact angle of a water droplet on these surfaces of ~55°, as measured in molecular dynamics simulations with the method employed by Werder et al.[69]. Periodic boundary

**Table 1 Slip length $\lambda$ as measured from MD.**

| Surface | Solvent | $\lambda$ (nm) |
|---|---|---|
| Graphene | Water | 60.0 ± 5.0 |
| Graphene | CPO | 40.0 ± 5.0 |
| Graphene | NMP | 13.0 ± 3.0 |
| GO (1%) | Water | 4.0 ± 2.0 |
| GO (2%) | Water | 2.0 ± 0.5 |
| GO (3%) | Water | 1.15 ± 0.2 |
| GO (4%) | Water | 1.07 ± 0.15 |
| GO (5%) | Water | 0.53 ± 0.12 |
| GO (6%) | Water | 0.37 ± 0.11 |
| GO (7.5%) | Water | 0.22 ± 0.11 |
| GO (9%) | Water | 0.09 ± 0.1 |
| GO (10.5%) | Water | 0.07 ± 0.1 |
| GO (12%) | Water | <0.05 |
| GO (13.5%) | Water | <0.05 |

GO stands for graphene-oxide and the percentage is the ratio between oxygen and carbon atoms.

conditions are used along the three orthogonal directions. Long-range Coulombic interactions were computed using the particle-particle particle-mesh (PPPM) method[70,71]. Fluid molecules are maintained at a constant temperature of 300 K with a Nosé-Hoover temperature thermostat[38,39] applied only to degrees of freedom in the $\hat{\boldsymbol{e}}_y$ and $\hat{\boldsymbol{e}}_z$ directions.

We used the TIP4P/2005 model for water[36] and the AMBER96 force field for the carbon-oxygen interactions, i.e., a Lennard-Jones potential with parameters $\epsilon_{CO} = 0.114$ kcal mol$^{-1}$ and $\sigma_{CO} = 0.328$ nm[37]. Water molecules are held rigid using the SHAKE algorithm[72]. A number $N = 10^4$ of water molecules is initially placed on a simple cubic lattice at the equilibrium density. The atmospheric pressure is imposed by imposing a normal force to the two moving walls. After ~10 ps, the moving walls reach their equilibrium position at $y = \pm H/2$, with $H \sim 8$ nm. Based on a preliminary convergence study, $H$ and the size of the computational box along $\hat{\boldsymbol{e}}_x$ were chosen large enough to avoid finite-size effect and to agree with the free space boundary condition approximation assumed in the BI simulations. After an equilibration phase of 20 ps, shear velocities of $u_s = 200$ m s$^{-1}$ and $u_s = -200$ m s$^{-1}$ are imposed on the top and bottom moving walls, respectively. This produces a shear rate $\dot{\gamma} \approx 5 \times 10^{10}$ s$^{-1}$. Typical shear rates in MD are usually a few orders of magnitude larger than the experimental ones. We verified to operate in the linear torque/shear rate response by exploring several different values of the shear rate. Due to the small size of the platelet, $a \sim 1.7$ nm, the Reynolds number is $\rho \dot{\gamma} a^2/\eta = 0.16$, for which the Stokes flow regime holds. Note that $\eta = 8.55 \times 10^{-4}$ Pa s for TIP4P/2005 water model[73]. After a second equilibrium phase of 20 ps, we recorded the total forces applied on the graphene atoms for 4 ns. To remove the hydrostatic contribution to the force on the graphene atoms, we performed a simulation in the absence of shearing ($u_s = 0$). We then subtract the force profile obtained in presence of shearing with the force profile obtained in absence of shearing.

Parameters of dynamic simulations are identical to the static case, except that the rigid graphene nanoplatelet is free to rotate around the $\hat{\boldsymbol{e}}_z$ axis, and free to translate in the $\hat{\boldsymbol{e}}_x$ and $\hat{\boldsymbol{e}}_y$ directions. The angle $\alpha$ of the platelet is recorder as function of the time. Ten independent simulations were performed, from which the average value of $\alpha$ as well as the standard deviation were calculated.

A graphene oxide platelet was generated by the addition of hydroxyl groups (−OH) to a monolayer graphene platelet of initial half length $a = 1.7$ Å. In order to build a realistic model of graphene oxide platelet, the reactive force field ReaxFF was used[74]. A total of four hydroxyl groups was added to each edge of the monolayer. When specified, a certain number of hydroxyl groups where also added to the basal plane at random location. A phase of relaxation at a temperature of 0 K allowed for the atoms to reach equilibrium positions. Finally, the final state of the graphene oxide platelet was frozen, and used as a rigid platelet for dynamic simulations in presence of solvent.

The atomic structure of no-slip platelet follows the calibrated surface and interaction parameters of a no-slip surface from Huang et al., with $\sigma_{ww} = 3.374$ Å, a close-packed density of $\rho_w = \sigma_{ww}^{-3}$, and $\epsilon_{ww} = 2.084$ kcal mol$^{-1}$[68]. The effective dimensions of the no-slip platelet are $a \approx 1.8$ nm and $b \approx 0.37$ nm.

Two non-aqueous solvents were considered, respectively $N$-Methyl-2-pyrrolidone (NMP) and cyclopentanone (CPO). The initial structure of NMP and CPO molecules is extracted from the automated topology builder[75]. We use the all-atom Gromos force field for NMP and CPO[76]. Graphene-NMP, graphene-CPO, wall-NMP and wall-CPO interaction parameters are calculated using the Lorentz-Berthelot mixing rules.

In order to measure the slip length from MD, we performed Poiseuille flow simulations of a liquid confined between two planes following the protocol by Herrero et al.[77]. In short, the position where the slip boundary condition applies is

determined from the Gibbs dividing plane, and the slip length is extracted from a fit of the Poiseuille flow profile in the bulk region. Slip length value for water, NMP and CPO are given in the Table 1. Note that the slip length value for water and pure graphene falls in the rather large range of values reported in the literature (between 1 and 80 nm[78]), and that the decrease of the slip length for increasing degree of oxidation is qualitatively consistent with results reported by Wei et al.[53]. Note also that in the case of multilayer graphene, the slip length along the edge of the platelet is expected to be smaller than on the basal plane due to the larger space between carbon atoms. Such space makes the potential energy landscape on the edges coarser than the potential energy landscape associated to the basal plane. A similar effect has been observed due to the intrusion of defect at the solid surface[42,79].

**Boundary integral formulation**. The boundary integral method is a computational method to solve the incompressible Stokes equation that requires a continuous surface onto which the integral equations are discretised. Examination of the molecular flow field indicates that a good approximation to the graphene surface is a rectangular parallelepiped with rounded edges (Fig. 1). The hydrodynamic thickness of a graphene platelet is set by the effective radius $\zeta$ of the carbon atoms, which is the radius as 'seen' by the water molecules. Here, $\zeta$ is fixed by the parameters of the Lennard-Jones potential between oxygen and carbon atoms (here $\sigma_{OC} = 3.28$ Å and $\epsilon_{OC} = 0.114$ kcal mol$^{-1}$), and is $\zeta \approx 2.5$ Å[33]. The projection of the approximated graphene surface on the $x - y$ plane can be parameterised as

$$h(s) = \begin{cases} b, & \text{if } |s| < a - \xi, \\ b\sqrt{1 - \left(\frac{|s|-a}{\xi}+1\right)^2}, & \text{otherwise.} \end{cases} \tag{8}$$

We denote this reference surface as $S$.

Owing to the small length scale of the graphene sheet in the MD system ($a \sim 1.7$ nm), the typical Reynolds number $\rho a^2 \dot{\gamma}/\eta \ll 1$ in the MD simulations, so the Stokes equations hold with very good approximation. In a boundary integral formulation, the incompressible Stokes equation is recast as an integral over the effective surface of the body[41]. We discretise the boundary integral equations on the reference surface $S$. In our case, the boundary integral equation for a point $\boldsymbol{x} \in S$ reads

$$\int_S \boldsymbol{n} \cdot \boldsymbol{K}(s', h') \cdot \boldsymbol{u}^{sl} dS - \frac{1}{\eta} \int_S \boldsymbol{G}(s', h') \cdot \boldsymbol{f} dS$$
$$= \frac{\boldsymbol{u}^{sl}(\boldsymbol{x})}{2} - \boldsymbol{u}_\infty(\boldsymbol{x}), \tag{9}$$

where $dS = \sqrt{||\partial_s \boldsymbol{x}||} ds_1$, $s' = s - s_1$, $h' = h(s) - h(s_1)$, $\boldsymbol{n}$ is the surface's normal and $\boldsymbol{G}$ and $\boldsymbol{K}$ are Green's functions corresponding to the 2D 'stokeslet' and 'stresslet', respectively[41]. The formulation in Eq. (9) accounts for a finite slip velocity at the boundary[23]; the slip velocity $\boldsymbol{u}^{sl}$ is closed in terms of the boundary traction $\boldsymbol{f}$ via the Navier slip boundary condition:

$$\boldsymbol{u}^{sl} = \frac{\lambda}{\eta} \boldsymbol{n} \times \boldsymbol{f} \times \boldsymbol{n}. \tag{10}$$

In the Supplemental Method 1, we decompose Eq. (9) into two independent scalar equations: one equation for $\Delta f_s$ and $\bar{f}_n$, and one equation for $\Delta f_n$ and $\bar{f}_s$. The asymptotic analysis and the numerical BI solutions are based on this latter formulation.

## Data availability

The data that support the findings of this study are available upon request to the authors.

## Code availability

All numerical codes in this paper are available upon request to the authors.

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

## Acknowledgements

We thank the European Research Council (ERC) for funding towards this project FLEXNANOFLOW (no. 715475). S.G. thanks support from FONDECYT's postdoctoral fellowships (no. 3170476). This research utilised Queen Mary's Apocrita HPC facility, supported by QMUL Research-IT, as well as the High Performance Computing infrastructure of the National Laboratory for High Performance Computing (NLHPC).

## Author contributions

C.K. wrote the article, performed and analysed the BI simulations, and developed the mathematical theory. S.G. wrote the article, and performed and analysed the MD simulations. L.B. designed the research, wrote the article, and analysed the results.

## Competing interests

The authors declare no competing interests.
