## [Peer Review File · Nature Communications]

Reviewers' comments:

Reviewer #1 (Remarks to the Author):

Review of NCOMMS-19-19129

The present manuscript proposes a numerical study of the motion of a nanoparticle in shear flow, accounting for the finite (and large) effective slip length of the fluid on the surface. It is motivated by the dynamics of graphene platelets, and finds that for large slip length the classical Jeffery orbits of an elongated solid in shear flow are not observed but instead the particle adopts a fixed and stable orientation with respect to the shear direction.

The findings are interesting. However, I do not believe that the manuscript meets the criteria for publication in a high impact scientific journal for the following reasons:

i) this manuscript is not the first one to have considered the question of the effect of Navier slip on the dynamics of elongated particles. For example, Zhang et al, PRE, 2015 have also considered this question and arrive to a different conclusion. Yet this work is not discussed and it would be necessary to understand whether the findings presented here are consistent and/or consider the same problem.

ii) the work is mostly numerical and considers two different types of simulations: (i) molecular dynamics simulations and (ii) boundary integral methods that solves the actual continuous Stokes equations. What is the motivation for using both methods? The authors spend a significant amount of time comparing the results, yet it is unclear why: one would expect them to give exactly the same thing. If it is not the case, shouldn't it be a matter of concern regarding the validity of MD simulations to represent Stokes' equations (or conversely)?

iii) Little physical insight is gained from the results. In contrast with experimental studies on the problem, the full flow field is accessible here but is not reported nor discussed. One would expect the authors to go beyond the simple reporting of the result (and the observation that one component of the force decreases more than another to lead to an analysis on the torque) but actually to explain why such variations of the force with the slip length are observed.

iv) The role of 2D vs. 3D analysis is not clearly addressed.

v) A particular geometry is chosen. Yet the authors note that the ends of the platelet contributes significantly to the dynamics. This seems to suggest that the actual precise geometry of the description is critical. Is it realistic to obtain therefore a good prediction knowing that in practice this information will be very difficult to obtain precisely?

Reviewer #2 (Remarks to the Author):

The paper explores an abnormal rotation of rigid graphene plates in a shear water because of the hydrodynamic slip at the solid-liquid interfaces. This interesting finding is first explored by molecular dynamics (MD) simulations and boundary integral (BI) methods, and then predicted from classical colloidal hydrodynamics. They discussed the mechanism of the stable orientation, and find that a stable orientation occurs when the hydrodynamic slip length is larger than the thickness of the platelet. While such a study is in principle interesting and could eventually deserve publication in Nature Communications, there are many points that should be reconsidered or clarified before this can be

recommended.

1. The authors have reported the inclination angle for graphene plates. The results would be highly affected by the rigidity of the graphene plates. For example, for a monolayer graphene, the planar structure would probably roll up or change to a rough plate, resisting the torque addressed by the shear flow. The bending rigidity has been discussed in the discussion section in the paper. They noted that graphene would behave as a rigid object at a certain condition. However, the condition does not work for the simulation systems in this work, due to the \sim nm size and $\sim 10^{10}$ shear rate. So the MD simulation work cannot be applied to support the main idea of this paper.

2. Could the author explain why does the torque of graphene change from positive to negative from a molecular view? Especially for $\alpha=0$, from figure 3, the torque reaches to the largest. Taking a monolayer graphene for example, the stress should be the same both on the upper and lower surfaces because of the negligible thickness. So where does the torque come from? The numerical solutions sure provide a clear understanding, but a physical view is needed.

3. Under such nanoscale studied in this paper, water molecules near graphene surfaces would not be likely continuous, for example, forming the layered structures. Thus, the stress on the graphene plates under a shear flow should be largely affected. In the paper the authors got consistent results for both MD and BI simulations. Do they suppose the density profile perpendicular to the graphene plane has little effect on the stress profile or torque? If so, it's better to demonstrate it with more details.

4. This paper studied the rotation of nanoparticles. It's better to show us the energy or torque profile under a full angle profile. Such as in figure 3, I recommend the authors showing the readers the torque profile at a angle ranging from 0° to 180° to avoid any misleading. This result would help find the other "hydrodynamic potential wells" proposed in this work.

5. In the discussion section, the authors extend their findings to other fluids such as NMP and ionic liquids. The slip lengths of these liquids on graphene surfaces are studied. However, the abnormal orientation would not happen when the graphene plates disperse in these liquids, due to the good dispersion. The slip between these liquids and graphene would not be found when the solid-liquid energy is larger than liquid-liquid. The authors are suggested carefully extending their findings.

6. The paper explores the rotation of graphene plates with different length and thickness. In a real colloid solution, would these kinds of plates exist, e.g., from liquid phase exfoliation? If they cannot affirm it, then the results in this paper would not be applicable in experiments or would be hard to realize.

7. The rotational dynamics of graphene demonstrated by MD simulations, however, would be convincing if the author can perform additional calculations for a no slip solids.

8. Other questions.

- (1) On page 2 the first paragraph, what does Jeffery's prediction refer to? Please also add a citation.
- (2) In the same paragraph, the authors raise a question that "It is currently unclear what effects may arise in graphene suspensions due to slip." Can they list some available phenomena in graphene suspensions resulted from the slip?
- (3) On page 2, "much larger than 1" should be "much larger than 1".
- (4) How did they calculate the rotational diffusion coefficient, D_r , since the plates were "constrained" by the shear flow? Is it calculated under equilibrium states?
- (5) On page 2, what is "the spanwise dimension of the computational domain ω "?
- (6) In figure 3, what is the unit of angle α ? Please also use only one unit through the manuscript.

Reviewer #3 (Remarks to the Author):

This manuscript addresses the role of hydrodynamic slip in the microhydrodynamics of particles. It is geared toward graphene nanoplatelets and their stacks. Particles suspended in a liquid and subject to flow are force and torque free, and as a consequence should move with and rotate with a rate equaling half the vorticity when they are spheres and under a no slip boundary condition. When they are non spherical and axisymmetric, so called Jeffery orbits are obtained, and when they are plate like the dynamics become more complicated. All this is relevant for processing as transport properties and final product properties depend on orientation. Here the effects of slip is investigated and it is shown that the particle can now satisfy the zero torque condition in different manner, not requiring rotation. this is a fundamentally new results, The particle being trapped in what the authors describe as a 'hydrodynamic potential well'. The results are intersting and novel. My only concern is the strong focus on graphene sheets, i list some concerns below. Other than that I think that new ideas have been generated in this paper, and found it very intersting and carried out with skill.

Recent studies indeed highlight the importance of hydrodynamic slip at the carbon-water interface, but this is done for solid interfaces or using implied using simulations. When graphene particles are to be suspended in an aqueous environment, their will typically be some oxidation required to make the graphene dispersable. For example, Mullens' methods to exfoliate graphene require an electrochemical oxidation, which leads to oxygen and hydroxyl groups which will be strongly hydrated and I doubt slip would be observed but am not familiar with the literature ions lip length in Graphene (i assume that the pacers cited were for pure graphene). Otherwise, particle may need to be functionalized, and then agains lip is unlikely (recent work by Natale and coworkers Langmuir 34 (26), 7844-7851 shows experimental data on functionalized graphene in non-aqueous environments which show Jeffery like behaviour). I hence believe that whenever you need to disperse particles in the aqueous medium, slip will be minimal if present at all. If the authors aims to keep the focus on graphene (which is not really necessary for the focus of the manuscript) this should be discussed. But it is not necessary at all. I wonder if some hydrogel particles, which could expel water and display slip layers would not be better examples.

1 Reviewer 1

The present manuscript proposes a numerical study of the motion of a nanoparticle in shear flow, accounting for the finite (and large) effective slip length of the fluid on the surface. It is motivated by the dynamics of graphene platelets, and finds that for large slip length the classical Jeffery orbits of an elongated solid in shear flow are not observed but instead the particle adopts a fixed and stable orientation with respect to the shear direction.

The findings are interesting. However, I do not believe that the manuscript meets the criteria for publication in a high impact scientific journal for the following reasons:

Authors' response: We strongly believe that our paper reports unexpected results on a subject that matters to a large and interdisciplinary audience, which should be of interest to the readers of Nature Communications. We hope that the comments below and the several amendments to the paper will provide a convincing argument in support of the innovative character and quality of the paper.

i) *this manuscript is not the first one to have considered the question of the effect of Navier slip on the dynamics of elongated particles. For example, Zhang et al, PRE, 2015 have also considered this question and arrive to a different conclusion. Yet this work is not discussed and it would be necessary to understand whether the findings presented here are consistent and/or consider the same problem.*

Authors' response: We are grateful to the reviewer for directing us to the article by Zhang et al. We have included this paper and others in the amended version of the paper. Below we explain why the paper by Zhang et al., although relevant, does not emphasise the same conclusions as ours because it focuses on a different parameter range. In the parameter range in which our papers overlap, the conclusions of the two papers are consistent. Furthermore, we have modified the manuscript to complete a discussion of the relevant the state of the art, and include references to the work of Zhang et al. and the papers listed above.

Zhang et al. restricted their study to slightly elongated particles (typically $a = 0.25$, $b = a/2$) with relatively small slip length ($\lambda = 0.1a < b$). In this parameter range, both the paper of Zhang et al. and ours predict a slip-dependent slowing down of the rotational dynamics. But our paper focuses primarily on the regime of relatively large slip length ($\lambda > b$), and very small aspect ratios ($b/a \ll 1$). This range was not discussed in Zhang et al.

In addition, in order to estimate the effective aspect ratio of a slightly elongated particle for larger slip length, Zhang et al. have made an extrapolation from the results they have obtained at small slip length. These results are based on the condition that the effective aspect ratio of the particle must tend to zero as the slip length tends to infinity. This condition is valid for an infinitely-long

cylinder with axis in the vorticity direction (see Supplementary Information of our paper) or a sphere [Luo et al. *J. Eng. Math*, 62, 1, 2008], but not for a thin platelet.

Apart from the work of Zhang et al., we have found one other article [Sellier *CMES*. 96, 159, (2013)] studying the effect of slip on the rotational dynamics of slightly-oblate or prolate particles held fixed in a linear shear flow with aspect ratio $a/b = 1.2$ and $a/b = 0.8$, and who study the effect of slip up to a maximum value of $\lambda/a = 1.0$. This article does not examine the hydrodynamic torque at different orientations of the particle, but for the oblate particle only examine the torque at one fixed angle, $\alpha = \pi/2$. This is not sufficient to understand the full dynamics of the particle, which requires a comparison at both $\alpha = \pi/2$ and $\alpha = 0$. Owing to the fact that they consider only slightly aspherical particles, these authors found, similarly to Zhang et al, a rather expected decrease in the hydrodynamic torque due to slip (they would have to consider a larger slip length $\lambda/a > 1$ to observe a change in sign of the torque).

There are some other papers which are pertinent, but their focus is different. We describe these papers below, and explain the difference with our case. When relevant, these papers are cited in the manuscript.

- Luo and Pozrikidis studied the motion of a spherical particle (rather than a plate-like particle as in our case) with Navier slip boundary conditions in a linear flow field near a wall [Luo and Pozrikidis, *J. Eng. Math*, 62, 1, 2008] and the interaction of two spheres with each other [Luo and Pozrikidis, *J. Fluid Mech*, 581, 129, 2007].
- Several articles, Sellier [*CMES*, 87, 157, 2012], Chang and Keh [*Theor. Comput. Fluid Dyn.* 26, 173, (2012)], Loyalka and Griffin [*J. Aerosol. Sci.* 25, 509, (1996)], Keg et al. [*Int. J. Eng. Sci.* 42, 371, (2004); *Int. J. Multiphase Flow*, 34, 713, (2008)] have studied the hydrodynamic drag resistance to rotation and translation for a spheroid with a slip surface. These results for the rotation are not directly applicable to our case: the torque produced by a freely rotating particle in a quiescent fluid is different from the torque produced by a particle held fixed in a linear shear flow.
- Other articles have focused on the rotational and translational drag produced by (i) spherical or nearly spherical particles i.e. [Basset, Vol 2. (1961); Leh and Chen, *Euro. J. Mech. B.Fluids*. 15,791, (1996); Chang and Keh, *J. Colloid. Int. Sci.* 330, 201 (2009)] or (ii) a torus [Loyalka *J. Aerosol. Sci.* 25, 371, (1996)] moving under rigid body motion with slip boundary conditions. Again, here the physics is different because the background flow is different and because of the very different geometry.

We agree with the reviewer that the paper should better acknowledge the excellent work done by others. To address the concern of the reviewer, we have modified the manuscript to complete a discussion of the relevant the state of the art, and include references to the work of Zhang et al and the papers listed above.

“In the context of colloidal hydrodynamics, slip is known to reduce the hydrodynamic stress applied by the shearing liquid on the particle’s walls, resulting e.g. in a slowing down of the rotational dynamics of spheres, and infinite cylinders with axis in the vorticity direction [Luo et al. J. Eng. Math, 62, 1, 2008, Kroupa et. al, Phys. Chem. Chem. Phys., 19, 5979, 2017] (Supplementary Information). A similar effect has been predicted for elongated particles of moderate aspect ratio ($b/a \sim 0.5$) and small slip length ($\lambda \sim a/10$) [Zhang et al., PRE, 91, 1, 2015], as well as for slightly oblate spheroids with $b/a = 5/6$, $\lambda/a \leq 1$, and their longer axis perpendicular to the plane of the flow [Sellier CMES. 96, 159, (2013)]. The effect of slip on the hydrodynamic torque and drag of rotating or settling elongated particles in quiescent fluid has also been studied [Loyalka and Griffin, J. Arerosl. Sci. 25, 509, (1996), Keh and Huang, Int. J. Eng. Sci. 42, 371, (2004); Keh and Chang, Int. J. Multiphase Flow, 34, 713, (2008)A. Sellier CMES, 87, 157, 2012 and Chang, Keh, Theor. Comput. Fluid Dyn. 26, 173, (2012)] and for plate-like geometries of relatively large thickness [Youngren and Acrivos, J. Chem. Phys. , 63, 3846 (1975)]. But graphene nanoplatelets can exhibit extreme aspect ratios (typically, $b/a \sim 10^{-3}$) and can have significant slip lengths, often larger than the nanoplatelet thickness. The effect of slip in such conditions must be reconsidered.”

“It could be expected that a small amount of slippage would just slow down the dynamics with respect to what is predicted by Jeffery’s theory such as in [Zhang et al., PRE, 91, 1, 2015]. Our result instead demonstrates that the presence of even relatively small slip can qualitatively change the rotational dynamics of the platelet by perturbing the balance of tangential and normal torques (Fig. 5).”

ii) *the work is mostly numerical and considers two different types of simulations: (i) molecular dynamics simulations and (ii) boundary integral methods that solves the actual continuous Stokes equations. What is the motivation for using both methods? The authors spend a significant amount of time comparing the results, yet it is unclear why: one would expect them to give exactly the same thing. If it is not the case, shouldn’t it be a matter of concern regarding the validity of MD simulations to represent Stokes’ equations (or conversely)?*

Authors’ response: The limits of a continuum treatment are not at all obvious in our case. The issue is not that of the validity of the continuum governing equations - the Navier-Stokes equation are essentially valid at the nanoscale [Bocquet and Tabeling, Lab on a Chip, 14, 17, 2014] - but that of the boundary conditions. While the adoption of a Navier slip boundary condition may be expected, without comparing against MD, the first-principle calculation of the slip length would not be possible. Nor it would be possible to prove that the use of a single slip length parameter is sufficient to capture the data for the

torque (as we have mentioned in the text, the slip lengths on the flat faces and edges of a graphene are in general different). Furthermore, unlike the continuum approach, MD captures effects such as fluid structuring near the solid, whose effects are a priori unknown.

A further motivation for combining continuum and atomistic approaches is the possibility of generalising the results to realistic length scales. MD simulations are intrinsically limited to relatively short nanosheets (length ≤ 5 nm), while realistic graphene sheets are typically in the micron range. By validating the continuum simulations in the nanoscale range, we have been able to develop a continuum model - devoid of free parameters - which we have used to explore realistic length and time scales.

A further point is that the continuum model is the basis of our asymptotic analysis. This analysis has enabled us to identify that the critical parameter which sets the dynamics is the ratio of slip length to platelet thickness. It would have not been possible to obtain this insight by performing MD only.

The combination of continuum and atomistic approaches has therefore been essential. In order to improve the manuscript's clarity on this point, we added the following sentence:

“The excellent agreement between the MD and BI calculations suggests that atomistic hydrodynamic features, such as fluid structuring near the solid-liquid interface [Tocci et al., Nanoletters, 14, 68726877, 2014] or non-uniformities in surface properties leading to differences in slip length between the edges and the flat surfaces [Joly et al, Physical Chemistry Letters, 7, 2016], do not induce essential effects on the torque. Therefore, a continuum formulation based on the solution of the Stokes equation with a single slip parameter can be used for predicting the stress applied on a graphene nanoplatelet.”

iii) *Little physical insight is gained from the results. In contrast with experimental studies on the problem, the full flow field is accessible here but is not reported nor discussed. One would expect the authors to go beyond the simple reporting of the result (and the observation that one component of the force decreases more than another to lead to an analysis on the torque) but actually to explain why such variations of the force with the slip length are observed.*

We accept that some aspects of the paper can be improved in clarity. But we respectfully disagree with the referee that the paper provides “little physical insight”. Some of the landmark papers on the microhydrodynamics of colloids in the low Reynolds number regime have been published without reference to any image of the flow field around the particle (see e.g. Jeffery, Proc. Roy. Soc. London, 102, 161 (1922); Bretherton J. Fluid. Mech, 14, 284 (1961); Singh et al. Phys. Fluids 26, 033303, 2014). The reason for this is simple: often the flow field distribution around the particle can be misleading, and the mapping between ambient flow field and hydrodynamic stress is in general not simple nor intuitive. This is particularly true in the low Reynolds number regime, in which

pressure and viscous stresses are all of the same characteristic magnitude and the flow disturbance long-ranged.

Our paper offers several aspects from which the reader can gain physical insights: numerics on the stress at both the atomistic and continuum level; a simple discussion on the physical origin of the stable orientation in terms of reduced shear stress at the flat faces of the platelet combined with independence of normal stresses on the slip length; asymptotics to make it clear when Jeffery’s results would hold and when they would not. Such combination of numerics with different approaches and mathematical theory with a rather sophisticated asymptotic technique goes beyond a mere reporting of the results.

All this considered, we thank the reviewer for pointing out that clarity and intuition about the flow is an aspect in which our initial manuscript could, evidently, be improved. To address the concern of the reviewer we have: i) included a discussion of the ambient velocity and pressure field in the Supplementary Information, see also Figure A1 below; ii) we have added a new figure (Fig. 5 in the new version of the manuscript) in order to better highlight the link between the platelet’s orientation and the hydrodynamic slip. See also the answer $n^{\circ}2$ to Reviewer 2.

Figure A1: Velocity field (arrows) and dimensionless pressure normalised by the shear stress $\eta\dot{\gamma}$ (colour) around a platelet with and without slip. The figure illustrates the flow field near a platelet aligned with the flow with slip (left) and without slip (right).

iv) *The role of 2D vs. 3D analysis is not clearly addressed.*

Authors’ response: To demonstrate the validity of our results when applied to a three-dimensional body, we have added a new section to the Supplementary Information. In the new section, we have extended the asymptotic analyses by Singh et al. for a 3D disk, and show that the total torque exerted on the 3D disk aligned with the flow also change sign for slip length above a critical slip length $\lambda_c \sim b$ [Singh et al. Phys. Fluids 26, 033303 (2014)].

In addition, we have performed finite element calculations (Comsol) of the

hydrodynamic torque acting on an ellipsoid of semi-axis $a = 1$ m (along x), $b = a/5$ (along y), and $c = a$ (along z), when the ellipsoid is exposed to a simple shear flow directed along the x axis). See the figure A2 below.

Figure A2: Ellipsoid used with the finite element calculations.

We have solved the Stokes equation using either no-slip or perfect slip boundary conditions. For shear rate $\dot{\gamma} = 0.1 \text{ s}^{-1}$ and viscosity of $1 \text{ Pa}\cdot\text{s}$, we find the following results using 2 different particle orientations (parameterised by α) and slip lengths λ . Confirming the agreement between the 2D and 3D results, the torque T changes sign only for a particle aligned with the flow with large slip:

	$\alpha = 0$	$\alpha = \pi/2$
$\lambda = 0$	$T = -0.096 \text{ N}\cdot\text{m}$	$T = -1.11 \text{ N}\cdot\text{m}$
$\lambda = \infty$	$T = 0.036 \text{ N}\cdot\text{m}$	$T = -1.14 \text{ N}\cdot\text{m}$

We also vary the depth parameter c for a particle with slip, and measure the ratio $-T(0)/T(\pi/2)$. The ratio converges toward a constant for c larger than a (Figure A3 below), showing that our 2D predictions are valid for particles that are sufficiently large in the z - direction . The following reference to the new section of the Supplementary Information has been added to the manuscript:

Note that the analysis of such quasi-2D configuration is not restrictive, and the results are valid for geometries that vary in the \vec{e}_z direction (e.g. a disk-like particle) up to a numerical prefactor (see the asymptotic analysis of the hydrodynamic traction for a 3D axisymmetric disk in the Supplementary Information).

This analysis can be repeated for objects of finite extent in the z - direction (e.g. a disk-shaped particle) and leads to similar results. The only difference is that one has to account for a numerical prefactor that depends on the specific geometry of the object (e.g. a circular disk vs a plate; see Supplementary Information).

v) *A particular geometry is chosen. Yet the authors note that the ends of the platelet contributes significantly to the dynamics. This seems to suggest that the actual precise geometry of the description is critical. Is it realistic to obtain therefore a good prediction knowing that in practice this information will be very difficult to obtain precisely?*

Figure A3: Ratio between torques $T(0)$ and $T(\pi/2)$ as a function of the depth parameter c in the case of perfect slip, as calculated from finite element simulations.

Author's response: We thank the reviewer to give us the opportunity to clarify this crucial aspect. In addition to platelets with a rectangular shape and rounded edges, we have also studied particles whose cross section is an ellipse (see the section II of the Supplementary Information). The asymptotic analysis of the hydrodynamic traction in the case of the ellipse leads to the same qualitative results as the one obtained with the quasi-rectangular platelets: the hydrodynamic slip induces a change in the sign of the torque for an orientation $\alpha = 0$. Therefore an ellipse with slip is expected to stabilise at a given angle with the flow (Supplementary Information). This result suggests that the specific shape of the particle is not important in dictating the essential elements of the particle dynamics. In order to clarify this point, we have added the following sentence to the manuscript:

Results have been obtained in the case of platelets with a rectangular shape and rounded edges (Fig 1 b), but an alignment is expected for a variety of plate-like shapes, such as for example a thin plate-like particle with an elliptical cross section (Supplementary Information).

Of course, considering different shape details will somewhat change the specific value of the stable angle, or make the relaxation time to equilibrium slightly smaller or larger, but this would not change the essential conclusions of the paper.

2 Reviewer 2

The paper explores an abnormal rotation of rigid graphene plates in a shear water because of the hydrodynamic slip at the solid-liquid interfaces. This interesting finding is first explored by molecular dynamics (MD) simulations and boundary integral (BI) methods, and then predicted from classical colloidal hydrodynamics. They discussed the mechanism of the stable orientation, and find that a stable orientation occurs when the hydrodynamic slip length is larger than the thickness of the platelet. While such a study is in principle interesting and could eventually deserve publication in Nature Communications, there are many points that should be reconsidered or clarified before this can be recommended.

We are glad that the reviewer find the results interesting. We hope that the amended version will address all the reviewer's concerns.

1. *The authors have reported the inclination angle for graphene plates. The results would be highly affected by the rigidity of the graphene plates. For example, for a monolayer graphene, the planar structure would probably roll up or change to a rough plate, resisting the torque addressed by the shear flow. The bending rigidity has been discussed in the discussion section in the paper. They noted that graphene would behave as a rigid object at a certain condition. However, the condition does not work for the simulation systems in this work, due to the $\sim nm$ size and $\sim 10^{10}$ shear rate. So the MD simulation work cannot be applied to support the main idea of this paper.*

Author's response: In fact the results will not change in character. We have carried out further MD simulations (see figure A4 below) with flexible nanoplatelets of the same size as those investigated in the original manuscript. The force field for the flexible graphene is AIREBO [Stuart, Journal of Chemical Physics, 112, 2000], which provides a realistic value of the bending rigidity ($B \sim 1.5$ eV). The platelet reaches a fixed angle with respect to the flow direction, as in the simulations with rigid graphene sheet.

Furthermore, we have carried out continuum simulations of a flexible sheet using the Boundary Integral method (see figure A5 below), for $(\eta\dot{\gamma}/D)(a/b)^2 \sim 1$. Also in this case we find that the nanoplatelet aligns with the shear flow. Only for $(\eta\dot{\gamma}/D)(a/b)^2 \gg 1$ the deformation of the sheets would be so large as to limit the validity of our results.

We can also offer more analysis. When the particle is aligned with the flow ($\alpha = 0$) the total normal viscous stress applied to the particle is

$$\sim \eta\dot{\gamma} [f_y(\alpha = 0) + (b/a)^2 f_y(\alpha = \pi/2)], \quad (1)$$

where f_y the local normal stress. The normal stress $f_y(\alpha = 0) \sim \mathcal{O}(b/a)$ over the surface, and $f_y(\alpha = 0) \sim \mathcal{O}(1)$ near the edges (in a region of size $\sim \mathcal{O}(b/a)$) (Fig. 4 of the main text). Also, $f_y(\alpha = \pi/2) \sim \mathcal{O}(1)$ (Supplemental Information). Hence, the normal viscous stress on a particle aligned with the

Figure A4: Snapshot of a flexible graphene nanoplatelet in a shear flow of water for different time instants. Platelet has dimensions $a = 2$ nm and $b = 0.25$ nm (monolayer $n = 1$), and the shear flow has a strength $\dot{\gamma} = 5 \cdot 10^{10} \text{ s}^{-1}$.

Figure A5: BI calculation for a (left) rigid platelet and a (right) flexible platelet, respectively.

flow as given by Eq. (1) is in fact $\sim \mathcal{O}(\eta \dot{\gamma} b a)$, much smaller than $\sim \mathcal{O}(\eta \dot{\gamma} a^2)$ which is expected for a particle facing the flow perpendicularly. In addition, we assume a cubic relationship of the Bending rigidity B with b , $B \approx D b^3$ for $D \approx 10^{11} \text{ J/m}^3$ [Lindahl et al., Nano Letters, 12, 7, 2012]. Hence equating the normal viscous stress with the bending forces $\sim B/a$, one expects buckling if

$$\eta \dot{\gamma} (a/b)^2 / D \sim 1. \quad (2)$$

For a platelets in high shear rate environment, as used in MD simulations, buckling is expected for $b/a < 10^{-2}$. For smaller shear rates, such as those

typically used for exfoliation, $\dot{\gamma} = 10^4 \text{ s}^{-1}$, and in water, buckling is expected for platelets with aspect ratio $b/a < 10^{-5}$. To improve the clarity of our manuscript on the assumption of rigid particle, we have modified the discussion section and included our scaling argument in a section in the Supplementary Information. The modified text in the discussion section of the manuscript reads:

“The results presented here are based on the assumptions that the nanoplatelet is rigid. For $\alpha = 0$, the platelet will behave as a rigid object provided that the viscous forces $\sim \eta\dot{\gamma}ab$ (Supplementary Information) are much smaller than the bending forces $\sim B/a$, where $B \sim Db^3$ is the bending rigidity [Poot et al, Appl. Phys. Lett. 92, 1], and $D \approx 10^{11} \text{ J/m}^3$ [Lingard, J. Colloid Interf. Sci. 49, 119, 1974 and Lindahl et al. Nano Lett. 12, 3526, 2012]. A criterion for the onset of deformability effects can be obtained by setting $\eta\dot{\gamma}(a/b)^2/D = 1$. For a typical shear rate $\dot{\gamma} = 10^4 \text{ s}^{-1}$, and a solvent with viscosity $\eta \sim 10^{-3} \text{ Pa}\cdot\text{s}$, a platelet with aspect ratio $b/a < 10^{-5}$ will appear rigid. For a 2 layers platelet, this corresponds to $a \sim 30 \text{ nm}$. For a 5 layers platelet, this corresponds to $a \sim 100 \text{ nm}$. Our criterion is conservative: the analogy with rods suggests that evident buckling accompanied by large deformations may appear for $\eta\dot{\gamma}(a/b)^2/D$ significantly larger than 1 [Becker and Shelley, Phys. Rev. Lett. 87, 198301 (2001)].”

2. *Could the author explain why does the torque of graphene change from positive to negative from a molecular view? Especially for alpha=0, from figure 3, the torque reaches to the largest. Taking a monolayer graphene for example, the stress should be the same both on the upper and lower surfaces because of the negligible thickness. So where does the torque come from? The numerical solutions sure provide a clear understanding, but a physical view is needed.*

Authors’ response: This is a delicate point, but one that we can explain in simple words. We consider a very small, but finite, value of b . Hence a clockwise torque component exists because the top surface is sheared from left to right, the bottom surface is sheared from right to left, and the lever arm is finite; from a molecular point of view the shearing force is due to the molecular momentum exchange at the solid-fluid interface. In addition there is a counter-clockwise torque component, due to the normal stresses (see fig A6 below); from a molecular standpoint this counter-clockwise torque component is essentially due to the fact that the molecules that hit the leading (left) edge of the particle have to make a sharp turn to satisfy the no-penetration condition at the edge, hence exerting a downward pressure on this edge, and similarly for the right edge. It turns out that when the no slip condition is satisfied exactly, the first torque component is larger than the second. But this is not the case if the slip length is larger than a threshold. The rotational trajectory is dictated by the difference between two torque components that are different by a very small, $O(b^2/a^2)$, amount, so our finding is highly non-trivial.

Furthermore, we want to point out that because of the boundary conditions, the presence in a flow of a nanoplatelet of vanishing thickness imposes a slip-dependent jump in the hydrodynamic stress between the two sides of the sheet. This would be true even in the case $b = 0$.

To comply with the reviewer’s suggestion, we have added a new figure to the manuscript (Figure 5 of the new version, see also fig A6 below), and have modified the text to clarify our explanations:

The observed dynamics can be understood from simple arguments, following a thorough analysis of Eq. 1. Let us call T_n the torque due to the normal traction (left term in Eq. 1), and T_s the torque due to the tangential traction (right term in Eq. 1). When the particle is aligned with the flow ($\alpha = 0^\circ$), because the tangential stress $h(s)\Delta f_s$ at the surface is reduced due to slip, T_s decreases by about one order of magnitude from the non-slip value (Fig. 4 (d)). But T_n in presence of slip decreases only by a factor of about 2 from the non-slip value (Fig. 4 (c)). This can be explained from the observation that the main contribution to T_n comes from the stress peaks near the edges; at the edges, the normal stress originates from the reorientation of the streamlines due to the non-penetrating boundary condition, and this effect is independent of λ . As a result of $T_n > T_s$, the total torque on the platelet for $\alpha = 0$ will become positive (counter clockwise) for sufficiently large slip length (Fig. 5). On the other hand, the direction of rotation when the particle is oriented normally to the flow ($\alpha = \pi/2$) is clockwise regardless of the value of λ (Fig. 5). Hence the particle finds an equilibrium orientation at an intermediate value of α .

Figure A6: (left) Vector plot of force distribution exerted by a shearing flow on a single-layer graphene particle aligned with the flow, as extracted from molecular dynamics. (right) Schematic of the dominant contributions to the torque applied on the platelet under shear flow for small slip length (a) and large slip length (b). The coloured arrows indicate the direction of rotation.

3. *Under such nanoscale studied in this paper, water molecules near graphene surfaces would not be likely continuous, for example, forming the layered structures. Thus, the stress on the graphene plates under a shear flow should be largely affected. In the paper the authors got consistent results for both MD and BI simulations. Do they suppose the density profile perpendicular to the graphene plane has little effect on the stress profile or torque? If so, it's better to demonstrate it with more details.*

Authors' response: The excellent agreement between BI and MD indeed suggests that the molecular structuring of water near the solid has only a very marginal effect on the rotational dynamics. Actually, one reason for our choice to carry out MD simulations was to test the validity of the continuum assumptions and to explore potential non-trivial layering effects (see also answer ii to Reviewer 1). It turns out that these effects, if at all present, are subdominant with respect to the effect of hydrodynamic slip.

Note that good agreement between continuum and molecular simulations for flows near graphene surfaces has been reported before, without the need to model layering effects. For example, a continuum description predicts *exactly* the permeability of water through carbon nanopores with diameter equal to the diameter of a water molecule [Gravelle et al., The Journal of Chemical Physics, 141, 18, 2014].

To make the points above more explicit in the paper, although succinctly, we have added the following sentence to the manuscript:

“The excellent agreement between the MD and BI calculations suggests that atomistic hydrodynamic features, such as fluid structuring near the surface [Tocci et al., Nanoletters, 14, 68726877, 2014], or the non-homogeneity of the slip length over the platelet surface [Joly et al, Physical Chemistry Letters, 7, 2016], do not induce leading order effects on the torque. Therefore, a continuum formulation based on the solution of the Stokes equation is relevant for predicting the stress applied on graphene nanoplatelet.”

4. *This paper studied the rotation of nanoparticles. It's better to show us the energy or torque profile under a full angle profile. Such as in figure 3, I recommend the authors showing the readers the torque profile at a angle ranging from 0 to 180 to avoid any misleading. This result would help find the other “hydrodynamic potential wells” proposed in this work.*

Authors' response: We have amended Fig. 3 of the main text to show the full profile. Thanks.

5. *In the discussion section, the authors extend their findings to other fluids such as NMP and ionic liquids. The slip lengths of these liquids on graphene surfaces are studied. However, the abnormal orientation would not happen when the graphene plates disperse in these liquids, due to the good dispersion. The slip between these liquids and graphene would not be found when the solid-liquid*

energy is larger than liquid-liquid. The authors are suggested carefully extending their findings.

Authors’ response: Our theory applies to well-dispersed systems, provided that the slip length is sufficiently large. NMP does provide good dispersion and, in the case of pure graphene, the slip length is sufficiently large for our theory to apply (of course assuming that the experiment is not done under conditions too far from those assumed in the theory). We have since performed multiple simulations of graphene nanoplatelet dispersed in NMP, and systematically observe a stable orientation (see the figure A7 below). See also the new section VI in the Supplementary Information, giving the orientation of a rigid platelet in NMP as a function of time.

Figure A7: Snapshot of a graphene nanoplatelet suspended in NMP, for different time instants. The AIREBO force field is used for graphene, and the All-Atom GROMOS force field for NMP [Schmid et al., European Biophysics Journal, 40, 843–856, 2011], the shear rate is of strength $\dot{\gamma} \sim 10^9 \text{ s}^{-1}$, studied nanoplatelet’s dimension are in the range $a = [1.5 - 2] \text{ nm}$ and $b/a = [0.1 - 0.3]$.

However, we agree with the reviewer that for very large shear rates, typically $\dot{\gamma} > 10^{10} \text{ s}^{-1}$, multilayered material could be exfoliated into single or few layer sheets. This effect would depend on the relation between solid-liquid and liquid-liquid surface energies (this precise relation is under investigation by many authors worldwide, although practical “rules of thumb” have been proposed by e.g. Coleman and coworkers). This question has to do with colloidal stability, not hydrodynamics per se. If the solvent is such that a multilayer particle becomes exfoliated under the operating flow conditions, the resulting single layer nanoplatelets would still be expected to obey our hydrodynamic predictions.

6. *The paper explores the rotation of graphene plates with different length and thickness. In a real colloid solution, would these kinds of plates exist, e.g., from liquid phase exfoliation? If they cannot affirm it, then the results in this paper would not be applicable in experiments or would be hard to realise.*

Author’s response: These kind of platelets should appear in experimental systems. The only requirement for a particle to align with the flow is that the slip length is larger than its thickness (we remind the reviewer that our theory suggests that the particle shape and particle lateral size are not important parameters). To consider a specific case, in liquid-phase exfoliation of graphite [Coleman, *Accounts Chem. Res.*, 46, 14–22 (2013)], a distribution of particle size is typically obtained. Assuming a typical slip length $\lambda \sim 10$ nm, only the particles thicker than about 10 nm would still rotate according to the classical Jeffery’s orbits (strictly speaking these particles are essentially ultra-fine graphite, not graphene). But the thinnest particles would obey the theory we propose. If the combination of solvent or the solid surface chemistry is such that the slip length is negligible, we expect of course the classical theory to hold.

7. *The rotational dynamics of graphene demonstrated by MD simulations, however, would be convincing if the author can perform additional calculations for a no slip solids.*

Authors’ response: We agree with the reviewer. We have now included such MD study for the no-slip case in the Supplementary Information (section V). We used a platelet of dimension $a \approx 1.7$ nm and $b/a = 0.37$, using the calibrated structure and interaction parameters of the no-slip surface from Huang et al. [Langmuir 24, 1442 (2008)]. Shear rate and box dimensions are equal to the values used in the manuscript for a slip platelet. As expected from both Jeffery’s theory [Jeffery, *Proc. R. Soc. Lond. A* 102, 161 (1922)] and BI calculations (blue line in Figure 3 of the manuscript), the no-slip platelet rotates continuously (see Figure A8 below), with a period equal to 0.45 ns, very close to that predicted by the continuum theory: $2\pi\dot{\gamma}^{-1}((b/a) + (b/a)^{-1}) = 0.52$ ns [Jeffery, *Proc. R. Soc. Lond. A* 102, 161 (1922)].

We have incorporated this new information by adding the following sentence to the manuscript:

“In contrast, MD simulations of a platelet presenting a no-slip surface produce orbits similar to those predicted by Jeffery (Supplementary Information).”.

8. *Other questions. (1) On page 2 the first paragraph, what does Jeffery’s prediction refer to? Please also add a citation.*

Authors’ response: “Jeffery’s prediction” means the particle rotates continuously. The text has been clarified as follows:

“In particular, we show that Jeffery’s theory [Jeffery, *Proc. R. Soc. Lond. A* 102, 161 (1922)], which predicts no stable orientation, fails to describe the rotational dynamics of graphene in the presence of comparatively large slip.”.

Figure A8: (left) MD snapshots for a no-slip platelet at four different times; (right) platelet orientation as a function of time.

(2) *In the same paragraph, the authors raise a question that “It is currently unclear what effects may arise in graphene suspensions due to slip.” Can they list some available phenomena in graphene suspensions resulted from the slip?*

Authors’ response: To the best of our knowledge, ours is the first prediction for the effect of slip suspensions of thin graphene particles in shear flow in the limit of slip lengths larger than the thickness. Slip is known to affect both rotational and translational drag coefficient for plate-like particles [Luo et al. J. Eng. Math, 62, 1, 2008]. This means an effect on the settling rate and on the rotational diffusion coefficient, for instance. We are not aware of other specific results for suspensions of graphene (or other thin plate-like particles) in shear flow that include considerations on the effect of slip. Reviewer 1 has made us aware of a publication by Zhang et al. focusing on slightly elongated particle in a shear flow. These authors have noticed that hydrodynamic slip lead to a reduction of the rotating velocity [Zhang et al., PRE, 91, 1-14, 2015], but their study is limited to comparatively small slip lengths and to slightly-elongated particle. See the answer i) to reviewer 1, as well as the new paragraph added to the manuscript (page 2).

(3) *On page 2, “much larger than 1” should be “much larger than 1”.*

Authors’ response: Corrected. Thanks.

(4) *How did they calculate the rotational diffusion coefficient, D_r , since the plates were “constrained” by the shear flow? Is it calculated under equilibrium states?*

Authors’ response: The rotational diffusion coefficient $D_r = k_B T / F_r$ for the rotary drag coefficient F_r has been estimated using the formula for a disk of radius a rotating in a quiescent fluid: $F_r \approx 32a^3\eta/3$ [Leal et. al. J. Fluid. Mech. 1971], where k_B is the Boltzmann constant, T the temperature and η

the fluid viscosity. We have since calculated this coefficient for a slip platelet (in 2D) in the limit of a platelet aspect ratio $b/a \rightarrow 0$. We found that, to leading order, D_r is independent of slip and that $F_r \approx 6.1a^3\eta$ (see Fig. A9 below). Our results demonstrate that the estimate for the disk gives the correct order of magnitude for the rotational diffusion coefficient of thin plate-like particles with slip.

Figure A9: Rotational drag coefficient F_r for a monolayer platelet and different slip length values λ . The red dashed line shows the value of F_r as $a/b \rightarrow \infty$.

(5) *On page 2, what is “the spanwise dimension of the computational domain w ”?*

Authors’ response: The spanwise dimension of the computational domain w is the size of the computational domain in the z direction (see the axes on Fig 1). The text has been clarified as follows:

The thickness of the platelet is $2b$, the length $2a$, and the spanwise dimension of the computational domain in the \vec{e}_z direction is w .

(6) *In figure 3, what is the unit of angle alpha? Please also use only one unit through the manuscript.*

Authors’ response: All axis are now in radians; the axis labels have been changed accordingly.

3 Reviewer 3

This manuscript addresses the role of hydrodynamic slip in the microhydrodynamics of particles. It is geared toward graphene nanoplatelets and their stacks.

Particles suspended in a liquid and subject to flow are force and torque free, and as a consequence should move with and rotate with a rate equaling half the vorticity when they are spheres and under a no slip boundary condition. When they are non spherical and axisymmetric, so called Jeffery orbits are obtained, and when they are plate like the dynamics become more complicated. All this is relevant for processing as transport properties and final product properties depend on orientation. Here the effects of slip is investigated and it is shown that the particle can now satisfy the zero torque condition in different manner, not requiring rotation. this is a fundamentally new results, The particle being trapped in what the authors describe as a 'hydrodynamic potential well'. The results are interesting and novel. My only concern is the strong focus on graphene sheets, i list some concerns below. Other than that I think that new ideas have been generated in this paper, and found it very interesting and carried out with skill.

Authors' response: We are grateful to the reviewer for recognising that the results are “fundamentally new”, “interesting” and “novel”. We understand the point of view of the reviewer, and provide new evidence in support of the applicability of our results. Generally, we would like to maintain the focus on graphene and other 2D nanomaterials because they are - by definition - extremely thin. So it is likely that slip lengths can be sufficient for our prediction of the stable orientation regime to apply. Furthermore, given the wide interest in solvents other than water, the specific concerns about dispersability of graphene in aqueous systems, although important, should not invalidate our results.

Recent studies indeed highlight the importance of hydrodynamic slip at the carbon-water interface, but this is done for solid interfaces or using implied using simulations. When graphene particles are to be suspended in an aqueous environment, their will typically be some oxidation required to make the graphene dispersable. For example, Mullens' methods to exfoliate graphene require an electrochemical oxidation, which leads to oxygen and hydroxyl groups which will be strongly hydrated and I doubt slip would be observed but am not familiar with the literature on slip length in graphene (I assume that the papers cited were for pure graphene). Otherwise, particle may need to be functionalized, and then again slip is unlikely (recent work by Natale and coworkers Langmuir 34 (26), 7844-7851 shows experimental data on functionalized graphene in non-aqueous environments which show Jeffery like behaviour). I hence believe that whenever you need to disperse particles in the aqueous medium, slip will be minimal if present at all. If the authors aims to keep the focus on graphene (which is not really necessary for the focus of the manuscript) this should be discussed. But it is not necessary at all. I wonder if some hydrogel particles, which could expel water and display slip layers would not be better examples.

Authors' response: The bulk of the paper, and all our initial studies, focus on graphene in water. This choice is methodological: we had to choose a system for MD, and the graphene-water system is the one for which practically

all trustworthy validation data for the interfacial hydrodynamic property is available. Other choices of solvent or nanomaterials as initial benchmarks would have led to methodological concerns. However, recognising the validity of the reviewer’s point of view, in the new version of the paper we have included: i) new simulation evidence that supports the applicability of our results to one solvent (NMP) in which graphene is dispersible; ii) a thorough discussion of the limitations of our results when some form of surface modification is applied to the particles, to caution the reader against the application of our results without critically assessing the experimental conditions. Furthermore, we show below preliminary MD results demonstrating stable orientation of hexabenzocoronene, a water-soluble macromolecule that is, geometrically, similar to a small graphene sheet (see figure A10).

Regarding the comparison with results of Natale et al. In view our our results, we believe it is too early to make strong statements about the validity of Jeffery’s predictions in predicting the full spectrum of rheological responses for nanometrically thin plate-like particles. For example, Natale et al. found that the average orientation angle of particles with respect to the steady shear flow does not go to 0 as the shear rate increases, but instead tends asymptotically toward a small but finite value. This feature is in disagreement with Jeffery prediction. While we don’t know yet if this behaviour is due to slip, we hope that the dissemination of our work may trigger a discussion on the effectiveness of Jeffery’s theory in predicting the full rheological features of graphene and similar plate-like nanoparticles.

Dispersion in NMP - Since the first submission, we have run new simulations to demonstrate that a stable orientation is obtained with N-Methyl-2-pyrrolidone (NMP), a very commonly solvent in which graphene is fully dispersible *without requiring alterations to the surface chemistry* [Hernandez et al., Nature nanotechnology, 3, 2008]. (Supplementary Information, see also the answer n°6 to reviewer 2).

Stable orientation of hexabenzocoronene, a water-soluble plate-like molecule Incidentally, our results are not limited to graphene and hold for any plate-like molecule as long as the hydrodynamics slip length is larger than the platelet thickness. For example, preliminary results show that our results apply to extended aromatic molecules, such as hexabenzocoronene of formula $C_{42}H_{18}$ (figure A10 below). These molecules are usually considered soluble in water [Englert et al., Chem. Commun., 46, 2010]. In keeping with our proposed theory, our MD simulations show that a $C_{42}H_{18}$ molecule aligns around a given angle α_c with the flow when exposed to a shear flow (figure A10 below). Again, we would like to maintain the focus on hydrodynamically smooth 2D nanomaterials in the paper as other macromolecules may have specificities which we are not a priori able to predict.

Modifications to the text - In addition to adding new results to the Supporting Information, we have modified the text as follows:

We have focused on graphene in water because of the quality and quantity of available data on the hydrodynamic properties of graphene-

Figure A10: (left) Snapshot of a $C_{42}H_{18}$ molecule, with carbon atoms in pink and hydrogen atoms in blue/green. (right) Angular distribution of α for a $C_{42}H_{18}$ molecule in a shear flow of strength $\dot{\gamma} = 100 \text{ ns}^{-1}$. The fluid is TIP4P/2005 water.

water interfaces, but the results are applicable to other solvents. Water is known not to be a good solvent for dispersing pure graphene, a limitation that can be circumvented by the chemical modification of the graphene surface [Stankovich et al, Carbon, 45, 1558–1565, 2007]. In the presence of surface modification, the validity of our prediction depends on the nature and extent of the surface modification. For example, oxidation at the graphene surface is known to reduce the slip length [Wei et al. Phys. Rev. E, 89, 1, 2014], and therefore under a large degree of oxidation, it is expected that Jeffery orbits should hold with good approximation. However, we expect the occurrence of a stable inclination when the oxidation is located exclusively at the graphene edges. Edge-selective oxidation makes graphene highly dispersible, while having little impact on its surface properties [Park et al. Nanoscale, 9, 1699–1708, 2017]. Alternatively, dispersion of graphene in water can be achieved by the addition of surfactants [Xu et al. Nanomaterials, 8, 2018]. Surfactants are expected to introduce non-trivial additional hydrodynamic effects, such as rate-dependent slip length [Zhu and Granick, Physical Review Letters, 87, 1–4, 2001]; their study is beyond the scope of the present article.

Graphene forms stable dispersion *without requiring alterations to the surface chemistry* in several solvents, for example in N-Methyl-2-pyrrolidone (NMP) [Hernandez et al. , Nat. nanotechnol, 3, 563 (2008)]. The slip length of NMP in contact with graphene is expected to be $\lambda = 12 \text{ nm}$ according to MD estimates we have made (Method section), and therefore our results should apply to graphene in NMP. Indeed, MD simulations of a graphene platelet suspended in a shear flow of NMP show that the platelet aligns at a small an-

gle, in qualitative agreement with the results obtained with water (Supplementary Information). Graphene also forms stable dispersion in certain Ionic Liquid (IL) [Ravula et al. *Nanoscale*, 7, 2015], and large slip length values have been reported for graphene in contact with IL ([mmIm^+][NTf_2^-]) [Voeltzel et al. *Journal of Physical Chemistry C*, 122, 2018]. A difference with the case studied in the current paper is that the slip length of IL on graphene strongly depends on the shearing velocity [Voeltzel et al. *Journal of Physical Chemistry C*, 122, 2018].

These results should apply to solid materials other than graphene, as long as the slip length is large enough. For example, the slip length of water on hexagonal boron nitride is predicted to be smaller than for graphene, about 3.3 nm [Tocci et al., *Nanoletters*, 14, 68726877, 2014], a value that is still roughly one order of magnitude larger than the typical thickness of a boron nitride nanosheet. Beyond 2D nanomaterials, large slip length have been reported on hydrogel surfaces [Kikuchi and Mochizuki, *Science and Technology*, 25, 2014],. Relatively large slip lengths can be obtained with conventional materials by using surface modification [Lee et al. *Phys. Rev. Lett.*, 101, 64501, 2008], depletion layers [Tuinier and Taniguchi, *J. Phys-Condens. Mat*, 17, L9, 2004; Fan, *Phys. Rev. E*, 75, 011803, 2007], or surface nanobubbles [Neto et al. ,*Rep. Prog. Phys.*, 68, 2859, 2005; Yang et al., *Europhys. Lett.*, 81, 64006, 2008], opening up opportunities for the experimental verification of our theory using mesoscale colloidal objects in simple or complex fluids.

Reviewers' comments:

Reviewer #1 (Remarks to the Author):

Review of manuscript NCOMMS-19-19129

I would like to thank the authors for taking the time to provide a detailed and constructed response to my earlier comments.

As emphasized in my earlier report, I do agree with the authors that the findings are interesting. In essence, it demonstrates the effect of the slip length on the tumbling dynamics of a thin solid in a shear flow. Yet I do not believe that it should be published in its present form in this journal for the following reasons:

i) I understand the authors argument regarding the necessity to validate the continuum approach with respect to molecular dynamics simulations. yet, while the continuum modeling approach (i.e. introducing a finite slip length in the boundary condition) is rather clear, how such an effect is accounted for in the MD simulations is not. In essence there should be a dimensional length scale hidden somewhere in the approach that plays a similar role.

ii) in line with the previous comment, the manuscript focuses on a specific application (graphene) with particular values of the dimensions of the system. While this is appreciable for an application point of view, the findings here should be general and the effects reported should be present, regardless of the particular system considered, provided the solid's thickness is small enough... But that latter point (small enough) only has a physical meaning when expressed in comparison with a reference scale (i.e. slip length). The use of dimensional units for presentation of the results therefore limits its generality.

iii) the authors have significantly expanded the Supplementary Information, which now contains much critical information. I believe such material is an integral part of the manuscript and should be included in the main text.

iv) the results are of great importance but in essence are better suited for a more specialized (e.g. fluid dynamics) journal. And I therefore do not believe that it should be published in Nature Communications but in a journal whose format provides more space to present their full analysis and implications.

Reviewer #2 (Remarks to the Author):

All of my comments and suggestions are well addressed. I recommend publication of this manuscript on Nature Communications, before the following minor revisions being addressed:

1. Can the authors discuss the rotation angles when the edges of the graphene plates are functionalized, e.g., by hydrogen or oxygenated group?
2. For the 5th comment, did the authors just fix the interlayer distances within the graphene plates in the MD simulations? If so, they would better describe it more clearly in the methods section.
3. On page 2, "therefore nor limited to water and graphene" should be "therefore not limited to water and graphene". On page 8, "The effective thickness of each graphene layer is fixed by the parameters of the Lennard-Jones potential between oxygen and carbon atoms", please revise it if there are mistakes.

Reviewer #3 (Remarks to the Author):

The authors have carefully addressed the reviewers' concerns and questions. I still find the observations interesting and the discussion stimulating. However, I remain concerned about the relevance for the large slip lengths when particles are in stable dispersed state.

When graphene particles are functionalized with a steric polymer brush, this brush will prevent slip, Likewise when oxide or hydroxyls are present hydration effects make slip unlikely and a discussion of the only experimental results which contradict the present calculations should be included with an even more clear discussion of when slip is expected.

Even for non-aqueous solvents where graphene is dispersible, the presence of ions has been suggested to explain the dispersibility. What the boundary condition on the particle will be is as yet unclear but it is unlikely to be as simple as being the large slip length for a large area surface.

I still find the paper interesting, but I am not yet convinced that the case of graphene flakes is the most relevant one. But then again the paper will for sure instigate discussion on this point. Maybe the authors could comment on which experimentally accessible parameters would be most worthwhile to pursue in light of feasibility and the power to discriminate the effects of slip

Responses to reviewer #1

I would like to thank the authors for taking the time to provide a detailed and constructed response to my earlier comments.

As emphasized in my earlier report, I do agree with the authors that the findings are interesting. In essence, it demonstrates the effect of the slip length on the tumbling dynamics of a thin solid in a shear flow. Yet I do not believe that it should be published in its present form in this journal for the following reasons:

i) I understand the authors argument regarding the necessity to validate the continuum approach with respect to molecular dynamics simulations. yet, while the continuum modeling approach (i.e. introducing a finite slip length in the boundary condition) is rather clear, how such an effect is accounted for in the MD simulations is not. In essence there should be a dimensional length scale hidden somewhere in the approach that plays a similar role.

Authors' response: In molecular dynamics, the value of the slip length λ is set by the atomistic interaction parameters. For water and graphene, λ mostly depends on the pair interaction energy ϵ_{OC} between carbon (C) and oxygen (O) atoms, which stems from the choice of force field model. In our case, using the TIP4P/2005 model for the water together with the AMBER96 force field for the graphene, we measure a slip length $\lambda = 60$ nm. Slip length measurements were made using a well-establish method [Herrero et al., J. Chem. Phys. (151) 2019], which consists in performing Poiseuille flow simulations of a liquid confined between two planes. The position where the slip boundary condition applies is determined from the Gibbs dividing plane, and the slip length is extracted from a fit of the Poiseuille flow profile in the bulk region. More details are given in the method section.

ii) in line with the previous comment, the manuscript focuses on a specific application (graphene) with particular values of the dimensions of the system. While this is appreciable for an application point of view, the findings here should be general and the effects reported should be present, regardless of the particular system considered, provided the solid's thickness is small enough. . . But that latter point (small enough) only has a physical meaning when expressed in comparison with a reference scale (i.e. slip length). The use of dimensional units for presentation of the results therefore limits its generality.

Authors' response: Dimensional units are only used in Figure 7 of the main text. In all other figures (including the new Figure 8), data are expressed in non-dimensional form. The use of dimensional units in some parts of the paper enables the reader to develop a better intuition of the parameter space in which our theory applies, which is particularly useful for experimentalist.

iii) the authors have significantly expanded the Supplementary Information,

which now contains much critical information. I believe such material is an integral part of the manuscript and should be included in the main text.

Authors' response: We agree with the reviewer that some of the parts previously presented in the Supplementary Information should be included in the main text. In the new version of the manuscript, we have moved all the results related to MD from the SI to the main text (see: the modified Figure 2; the new section named 'Dispersability and surface modification'; and the new Figure 8). Technical details have been moved to the Methods section (see subsections 'Surface modification', 'No-slip platelet' and 'Non-aqueous solvents'). The new version of the SI only contains details about the Boundary Integral calculations (these details have not been moved to the Methods section because of the lengthy equations involved in their analysis), some details about the rigidity assumption, and an example of the flow field.

iv) the results are of great importance but in essence are better suited for a more specialized (e.g. fluid dynamics) journal. And I therefore do not believe that it should be published in Nature Communications but in a journal whose format provides more space to present their full analysis and implications.

Authors' response: We thank the reviewer for pointing out that the 'results are of great importance'. It is indeed for this reason that we have opted for Nature Communications, and not a specialised fluid dynamics journal. The results should be of interest to a broad scientific audience, that includes materials scientists, rheologists, and people interested in the physico-chemical aspects of colloidal systems.

According to Nature Communications website, 'papers published by the journal [Nature Communications] represent important advances of significance to specialists within each field', and we note that several important results of fluid dynamics and applied mathematics have been published in this journal. Recent examples include: Goyette et. al, *Microfluidic multipoles theory and applications*, Nat. Comms., 10, 2019; Gao et. al, *Active control of viscous fingering using electric fields*, Nat. Comms., 10, 2019; Broeren et. al, *A general method for the creation of dilational surfaces*, Nat. Comms., 10, 2019.

We are currently preparing a second manuscript, to be submitted to a specialised fluid dynamics journal, where we explore in detail the mathematical aspects of the problem described in the current manuscript.

Responses to reviewer #2

All of my comments and suggestions are well addressed. I recommend publication of this manuscript on Nature Communications, before the following minor revisions being addressed:

1. *Can the authors discuss the rotation angles when the edges of the graphene plates are functionalized, e.g., by hydrogen or oxygenated group?*

Author’s response: We have added a new section named ‘Dispersability and surface modification’ to the main text, as well as the new Figure 8 (reported above as Figure A1). This figure shows that the time-averaged angle of the graphene plate with edge-selective oxidation is similar to the angle obtained with a pure graphene platelet of same size. This result is discussed in the main text of the new version of the manuscript (page 6). We report the text here:

‘To illustrate the effect of edge-selective modification on the orientation of a graphene nanoplatelet in water, we have performed additional MD simulations using a monolayer graphene platelet presenting edge-selective oxidation (Methods). The results show no significant change in the time-average equilibrium angle α_c as compared with pure graphene (Fig. 8 (a)). This is inline with our theory which shows that the contribution to the torque from the edges is independent of λ to leading order. Thus, the validity of our predictions is not undermined by edge modification, and our theory should apply to cases in which particle aggregation is prevented by modification of the edges (including inducing charges at these locations [Li et al., Nat. Nanotechnol., 3, 101 (2008)] [Yi et al., Chem. Commun., 49, 11059 (2013)].’

2. *For the 5th comment, did the authors just fix the interlayer distances within the graphene plates in the MD simulations? If so, they would better describe it more clearly in the methods section.*

Author’s response: In our answer to the 5th comment, the simulations were made using a flexible force field named Airebo [Stuart et al., J. Chem. Phys., (112), 2000], which is different from the rigid force field used in the present manuscript. With a flexible force field, the interlayer distance between graphene layers is re-calculated at each time-step by the molecular dynamics algorithm. The resulting equilibrium distance corresponds to the distance that minimize the energy of the system, and is mostly set by the carbon-carbon Lennard-Jones parameters. However, in the case of a rigid model, such as the one used in the present manuscript, the distance was chosen to match the experimental value. We have added the following clarification to the Method section:

In the case of a multilayer graphene platelet, the distance between two layers of graphene is chosen to be equal to the experimental

Figure A1: (A) Average angle α_c as a function of λ/b for monolayer graphene with $a = 1.7$ nm. Data corresponds to pure graphene in water (open disk), pure graphene in NMP (open square), pure graphene in CPO (open triangle), and surface modified graphene in water (full disks). (B) Average period P of the rotational orbit, as a function of λ/b . Data corresponds to a no-slip platelet in water (open disk), no-slip platelet in NMP (open square), no-slip platelet in CPO (open triangle), and surface modified graphene in water (full disks).

value (3.35 Å [Chung, Review Graphite, J. Mater. Sci., 37, 2002]).

3. On page 2, “therefore nor limited to water and graphene” should be “therefore not limited to water and graphene”.

Author’s response: Corrected. Thanks.

On page 8, “The effective thickness of each graphene layer is fixed by the parameters of the Lennard-Jones potential between oxygen and carbon atoms”, please revise it if there are mistakes.

Author’s response: We agree that this sentence was not completely clear. We have added a new subsection to the Methods called ‘Reference surface’:

The boundary integral method is a computational method to solve the incompressible Stokes equation that requires a continuous surface onto which the integral equations are discretised. Examination of the molecular flow field indicates that a good approximation to the graphene surface is a rectangular parallelepiped with rounded edges (Fig. 1). The hydrodynamic thickness of a graphene platelet is set by the effective radius ζ of the carbon atoms, which is the radius as ‘seen’ by the water molecules. Here, ζ is fixed by the parameters of the Lennard-Jones potential between oxygen and carbon atoms (here $\sigma_{OC} = 3.28\text{Å}$ and $\epsilon_{OC} = 0.114\text{kcal/mol}$), and is about $\zeta \approx 2.5\text{Å}$ [Gravelle et al., J. Chem. Phys, 141, 18C526, (2014)].

The projection of the approximated graphene surface on the $x - y$ plane can be parameterised as

$$h(s) = \begin{cases} b, & \text{if } |s| < a - \xi, \\ b\sqrt{1 - \left(\frac{|s| - a}{\xi} + 1\right)^2}, & \text{otherwise.} \end{cases}$$

We denote this reference surface as S .

Responses to reviewer #3

The authors have carefully addressed the reviewers' concerns and questions. I still find the observations interesting and the discussion stimulating. However, I remain concerned about the relevance for the large slip lengths when particles are in stable dispersed state. (...)

Author's response: We understand the point of view of the reviewer, but in the new version of the manuscript we emphasise that there are surface modification methods that lead to a well dispersed system without affecting their surface slip properties significantly. For example, good dispersion of graphene water can be achieved by oxidation of the edges of the particles rather than of their basal plane [Park et al. *Nanoscale*. 9, 1699 (2017)] [Aliyeva et al. *Soft Mater*. 17, 488, (2019)]. Since the hydrodynamic traction distribution at the edges does not depend on the slip length to leading order (as shown by our mathematical analysis), in the presence of a chemical modification that only affects the edges but not the graphene's basal plane our theory must apply. To convince the reader of this, we have carried out new MD simulations with edge-oxidatised graphene particles suspended in water. These simulations show the emergence of a stable angle, and comparisons with pure graphene of same dimension confirm that the equilibrium angle is hardly affected by selective edge oxidation.

To address the reviewer's comment, we have added a new section to the manuscript called 'Dispersability and surface modification' (starting page 6), see also the new Figure 8, included in these notes as Figure A1. Concerning edge-selective modification, the new section writes:

'To illustrate the effect of edge-selective modification on the orientation of a graphene nanoplatelet in water, we have performed additional MD simulations using a monolayer graphene platelet presenting edge-selective oxidation (Methods). The results show no significant change in the time-average equilibrium angle α_c as compared with pure graphene (Fig. 8 (a)). This is inline with our theory which shows that the contribution to the torque from the edges is independent of λ to leading order.'

(...) When graphene particles are functionalized with a steric polymer brush, this brush will prevent slip, Likewise when oxide or hydroxyls are present hydration effects make slip unlikely.. (...)

Author's response: It is true that a surface modification of the basal plane will in most practical cases reduce the slip length. We discuss this issue in the new version of the manuscript. In particular, we have carried out simulations in which hydroxyl groups ($-\text{OH}$) were added to the basal plane of a graphene platelet at random locations. Our results show that, for an oxygen/carbon fraction approximately less than 6%, a stable orientation occurs, as predicted by

our theory (see the new Figure 8, reported above as Figure A1). The crucial question is therefore whether surface functionalisation will keep the slip length within the “stable angle” regime (i.e. approximately larger than the particle half-width). The quantification of the effect of surface chemistry and structure on surface slip for different 2D nanomaterial/solvent combination is an important aspect, but one which is separate from the main message of the paper. New numerical and experimental tools, that are just now being developed (see e.g. Tocci et al., Nano Letters (14), 2014, and the experimental work by John Sader’s group, University of Melbourne), will be necessary to obtain a complete picture of the effect of functionalisation on slip.

To address the reviewer’s comment, we have described the new simulations and the corresponding literature information in the new section ‘Dispersability and surface modification.’ (starting page 6), and added the new Figure 8. We have also added the following text to the discussion (page 7):

‘The effect of surface modification of the basal plane, however, must be assessed more critically. We showed that an increase in the degree of oxidation at the basal plane eventually alters the stable orientation and leads to a transition toward continuum rotation of the particle (Fig. 8), in agreement with our theory which predicts a transition for $\lambda \sim b$.’

(...) a discussion of the only experimental results which contradict the present calculations should be included with an even more clear discussion of when slip is expected.

Author’s response: We agree with the reviewer that the sentence (from the previous manuscript)

‘For example, oxidation at the graphene surface is known to reduce the slip length [Wei et al., Phys. Rev. E, 89, 1 (2014)], and therefore under a large degree of oxidation, it is expected that Jeffery orbits should hold with good approximation.’

does not provide sufficient details on the effect of surface modification on the particle dynamics. The paper by Wei et al. is actually based on MD simulations. We have since reproduced their slip length measurements for varying degree of graphene oxidation (see the new table I of the manuscript, and the new ‘Surface modification’ subsection in the Methods section). Our results show that the slip length remains larger than 0.25 nm (i.e. the half width of a monolayer graphene) as long as the oxygen/carbon ratio $r_{C/O}$ remains below 6%, approximately. Accordingly, a stable orientation of monolayer platelets is obtained for $r_{C/O} < 6\%$, in agreement with our theory (see the new Figure 8 and the new section ‘Dispersability and surface modification.’). For further details, see also the response to Reviewer 2, question 1.

Even for non-aqueous solvents where graphene is dispersible, the presence of ions has been suggested to explain the dispersibility. What the boundary condition on the particle will be is as yet unclear but it is unlikely to be as simple as being the large slip length for a large area surface.

Author’s response: We agree with the reviewer, adsorbed ions at the graphene surface are known to improve dispersion stabilisation because of Coulomb repulsion [Smith et al., New J. Phys., 12 (2010)]. A non-zero surface charge may reduce the slip length, with consequences for the applicability of our theory [Pan and Bhushan, J. Colloid Inter. Sci., 392 (2013)]. However, nanometric slip length and non-zero surface charge are not mutually exclusive, as highlighted by detailed flow measurements in nanotubes [Secchi et al., Nature, 537 (2016)]. In the case of carbon nanotubes, slip lengths in the range of 17 – 300 nm have been reported by Secchi et al. for a surface charge $\sim 5 \text{ mC/m}^2$, and for boron-nitride nanotubes, slip lengths of $\lesssim 5 \text{ nm}$ and surface charge $\sim 100 \text{ mC/m}^2$.

I still find the paper interesting, but I am not yet convinced that the case of graphene flakes is the most relevant one. But then again the paper will for sure instigate discussion on this point.

Author’s response: As stated in our original manuscript, we initially focused on graphene nanoplatelets in water because of the quality and quantity of available data on the hydrodynamic properties of graphene-water interfaces. Our results, however are much more general than this specific case, and perhaps this generalisation was not sufficiently well discussed in the previous version of the manuscript.

To address the reviewer’s comment, we have added a whole new sub-section, entitled ‘Dispersability and surface modification’. This sub-section explains in detail in which experimental cases our theory should apply, discusses issues of dispersability comparing edge-modified surfaces to basal-plane modified surfaces, and discusses the general applicability of the results beyond aqueous dispersions and graphene. In this sub-section we have included our MD simulations of graphene dispersed in NMP (which were previously placed in the Supplementary Information), and discuss the new MD simulations on the effect of oxidation (discussed above). This new information should convince the reader that stable dispersions of graphene with significant slip could be obtained with controlled functionalisation and/or through the use of suitable solvents.

Maybe the authors could comment on which experimentally accessible parameters would be most worthwhile to pursue in light of feasibility and the power to discriminate the effects of slip

Author’s response: We have carried out further theoretical analysis and have identified two experimentally accessible parameters that could be used to provide experimental evidence in support of the validity of our theory: the so-called

Figure A2: Comparison of our theory (solid and dashed lines) with experimental results (symbols) from [Reddy et al. Langmuir, 34, 7844, 2018] for χ (left) and $\Delta n'' / (\Delta n''_{\max})$ (right). FGS stands for functionalised graphene platelets, and data are compared for $k_e \approx 0.016$ (no-slip) and $k_e \approx 0.035i$ (slip with $\lambda = 20$ nm). NP stands for slightly oblate nano-spheroids, with $b/a = 0.55$ and lengths 170 nm and 290 nm, and data are compared for $k_e \approx 0.55$ (no-slip) and $k_e \approx 0.46$ (slip with $\lambda = 20$ nm).

“degree of orientation” parameter [Frattoni & Fuller, J. Fluid. Mech., 168, 119 (1986)] and the effective suspension viscosity.

Following the previous comment of the reviewer concerning the article by Natale et al. on Phys. Rev. Fluids, we have compared our prediction to linear dichroism data from Natale and collaborators [Natale et al. Phys. Rev. Fluids, 36, 963393, 2018 and Reddy et al. Langmuir, 34, 7844, 2018]. We found that slip does affect the average orientation angle χ at large Pe (Fig. A2, left). However a more sensitive measure of orientation is the “degree of orientation” parameter $\Delta n'' / \Delta n''_{\max}$ (defined in [Frattoni & Fuller, J. Fluid. Mech., 168, 119 (1986)]). When comparing $\Delta n'' / \Delta n''_{\max}$ for slightly spheroidal particles of lengths 170 nm and 292 nm, our theory shows that the effect of slip on $\Delta n'' / \Delta n''_{\max}$ is quite significant and should be measurable (Fig. A2, right).

The second experimentally accessible parameter that could be used to quantify the effect of hydrodynamics slip is the steady-state effective viscosity. We have calculated the effective viscosity of a dilute suspension of nanoplatelets with and without slip, by evaluating via the boundary integral formulation the particle contribution σ'_{xy} to the bulk shear stress in a suspension of nanoplatelets subject to a macroscopic simple shear flow in the $x - y$ plane. Our results, soon to be submitted to a leading fluid dynamics journal, show that the particle stress σ'_{xy} at large Pe numbers can be reduced by up to a factor 4 for a slip length

Figure A3: A comparison of the dimensionless particle stress $\sigma_{xy} = \sigma'_{xy}/(c\eta\dot{\gamma})$ versus platelets length for a dilute suspension of nanoplatelets presenting surfaces with different slip lengths. The calculation is done for $Pe \rightarrow \infty$.

$\lambda = 2$ nm (Fig. A3). The corresponding large reduction in viscosity should be measurable with well-calibrated equipment under controlled conditions.

To address the reviewer’s suggestion in the manuscript, in the discussion section we have commented upon the possible use of the above experimentally accessible parameters, adding the following text:

‘Such decrease in viscosity, which could be relevant for improving the flowability of graphene inks, could be measured experimentally as a way to evidence slip effects (a similar suggestion was made in [Kroupa et al., Phys. Chem. Chem. Phys., 19, 5979 (2017)] for concentrated dispersions of spherical particles).’

‘The validity of the theory discussed here could be tested by measuring experimental observables that are sensitive to second-order statistical moments, such as the ‘degree of orientation’. The ‘degree of orientation’ of the particles, can be assessed by rheo-optics experiments [Frattini and Fuller, J. Fluid Mech., 168, 119 (1986)], [Natale et al., Phys. Rev. Fluids., 3, 063303 (2018)], [Reddy et al., Langmuir, 34, 7844 (2018)]. Contrarily, the average particles orientation angle may not be ideally suitable for discriminating between rotating and aligned particles because highly-elongated plate-like particles are expected to align with the flow in a time-average sense regardless of the hydrodynamic slip [Leal & Hinch, J. Fluid Mech., 46, 685 (1971)] [Hinch & Leal, J. Fluid Mech., 52, 683 (1972)].’

REVIEWERS' COMMENTS:

Reviewer #1 (Remarks to the Author):

I appreciate the authors taking the time to revise their manuscript and responding to my earlier comments.

This is now the third time that I have reviewed this manuscript. As emphasised in previous iterations, I do believe that the results and analysis presented by the manuscript are of important interest to the fluid dynamics community as they unveil a particular fundamental effect associated with hydrodynamic slip. I do maintain that I still have doubt regarding the impact of such results on a broader community and still recommends publication in a more technical journal (but not the duplication of results in a subsequent publication in such a journal). Yet, at this point, I believe that the authors and I do have different points of view on this, and that it is a matter of editorial decision.

Should the manuscript be recommended for publication in Nat. Comm., I however ask the authors to address the following minor elements prior to publication:

- a critical element in this problem is obviously the ratio of the effective slip λ and the thickness of the platelet. While I understand that the use of dimensional quantities is more adapted for this type of publication and to attract a broader audience, I am afraid that the importance of this ratio is somewhat relegated toward the end of the manuscript while it is of critical importance. For example, on Figure 3, it should be clearly understood that the difference between the three curves is essentially the slip-to-width ratio (with the no-slip condition corresponding to $\lambda \ll \text{thickness}$) and the two other curves differing by a factor 2 in that ratio). This is difficult to grasp as the dimensional values for λ , a and b are not indicated and only the value of b/a is given (which is much less of a physically-relevant parameter to understand what is happening here).
- the authors should discuss directly in the manuscript that the value of the slip length is not an a priori fixed parameter of the MD simulation but instead a product of it, and may therefore depend on the flow configuration considered (to measure it in the Methods section, it is necessary to consider a particular example).

Reviewer #2 (Remarks to the Author):

All of my comments and suggestions are well addressed. It is my pleasure to recommend it for publication now.

Reviewer #3 (Remarks to the Author):

The Authors have taken considerable time and effort to address the issue of what the slip length will be for a dispersed particle, with the necessity to have other oxide groups or steric stabilizers. The assumptions made by the authors make it clear that the issue slip in a real dispersion will be very subtly dependent on where e.g; the ionic groups will be located on the disks. The authors cleverly identify ways in which slip on particles could occur and nicely quantify the slip length and in particular quantified this.

So in conclusion I still found the theoretical analysis of interest to the community of those doing fluid mechanics and suspension rheology, but I remain somewhat skeptical of forcing these ideas onto graphene dispersions. There is simply no way to directly evaluate if the assumptions which were made

here - almost to ensure that slip is still going to be present, if these correspond to those conditions realized in a real physical experiment.

Responses to Reviewer 1

I appreciate the authors taking the time to revise their manuscript and responding to my earlier comments. This is now the third time that I have reviewed this manuscript. As emphasised in previous iterations, I do believe that the results and analysis presented by the manuscript are of important interest to the fluid dynamics community as they unveil a particular fundamental effect associated with hydrodynamic slip. I do maintain that I still have doubt regarding the impact of such results on a broader community and still recommends publication in a more technical journal (but not the duplication of results in a subsequent publication in such a journal). Yet, at this point, I believe that the authors and I do have different points of view on this, and that it is a matter of editorial decision.

Should the manuscript be recommended for publication in Nat. Comm., I however ask the authors to address the following minor elements prior to publication:

A critical element in this problem is obviously the ratio of the effective slip λ and the thickness of the platelet. While I understand that the use of dimensional quantities is more adapted for this type of publication and to attract a broader audience, I am afraid that the importance of this ratio is somewhat relegated toward the end of the manuscript while it is of critical importance. For example, on Figure 3, it should be clearly understood that the difference between the three curves is essentially the slip-to-width ratio (with the no-slip condition corresponding to $\lambda \ll \text{thickness}$) and the two other curves differing by a factor 2 in that ratio). This is difficult to grasp as the dimensional values for λ , a and b are not indicated and only the value of b/a is given (which is much less of a physically-relevant parameter to understand what is happening here).

Authors answer: We thank Reviewer 1 for the suggestion. We have included the half length and aspect ratio of the platelets, as well as estimated slip length in the captions of Fig 2 and Fig 3.

The authors should discuss directly in the manuscript that the value of the slip length is not an a priori fixed parameter of the MD simulation but instead a product of it, and may therefore depend on the flow configuration considered (to measure it in the Methods section, it is necessary to consider a particular example).

Authors answer: We thank Reviewer 1 for the suggestion, the following sentence has been added to the manuscript:

The slip length of water on a planar graphene surface, which depends on the force fields, is estimated here from Poiseuille flow simulations as $\lambda = (60 \pm 5)$ nm (see Methods).

Responses to Reviewer 3

The Authors have taken considerable time and effort to address the issue of what the slip length will be for a dispersed particle, with the necessity to have other oxide groups or steric stabilizers. The assumptions made by the authors make it clear that the issue slip in a real dispersion will be very subtly dependent on where e.g; the ionic groups will be located on the disks. The authors cleverly identify ways in which steric groups on particles could occur and nicely quantify the slip length and in particular quantified this.

Authors answer: We thank Reviewer 3 for the positive comments on the manuscript.

So in conclusion I still found the theoretical analysis of interest to the community of those doing fluid mechanics and suspension rheology, but I remain somewhat skeptical of forcing these ideas onto graphene dispersions. There is simply no way to directly evaluate if the assumptions which were made here - almost to ensure that slip is still going to be present, if these correspond to those conditions realized in a real physical experiment.

Authors answer: We agree with Reviewer 3 that focusing on graphene, as we did in the previous versions of the manuscript, is not necessary. In order to address Reviewer 3's concern, and following the recommendations of the Editor, we have refocused the manuscript to nanoplatelets suspended in a shear flow instead of graphene. The title of the manuscript has been changed into '*Hydrodynamic slip can align thin nanoplatelets in a shear flow*', and the abstract, introduction, and discussion have been amended to reflect the new focus of the article.